# EurNet: Efficient Multi-Range Relational Modeling of Spatial Multi-Relational Data

## Abstract

Modeling spatial relationship in the data remains critical across many different tasks, such as image classification, semantic segmentation and protein structure understanding. Previous works often use a unified solution like relative positional encoding. However, there exists different kinds of spatial relations, including short-range, medium-range and long-range relations, and modeling them separately can better capture the focus of different tasks on the multi-range relations (*e.g.*, short-range relations can be important in instance segmentation, while long-range relations should be upweighted for semantic segmentation). In this work, we introduce the **EurNet** for **E**fficient m**u**lti-range **r**elational modeling. EurNet constructs the multi-relational graph, where each type of edge corresponds to short-, medium- or long-range spatial interactions. In the constructed graph, EurNet adopts a novel modeling layer, called *gated relational message passing* (**GRMP**), to propagate multi-relational information across the data. GRMP captures multiple relations within the data with little extra computational cost. We study EurNets in two important domains for image and protein structure modeling. Extensive experiments on ImageNet classification, COCO object detection and ADE20K semantic segmentation verify the gains of EurNet over the previous SoTA FocalNet. On the EC and GO protein function prediction benchmarks, EurNet consistently surpasses the previous SoTA GearNet. Our results demonstrate the strength of EurNets on modeling spatial multi-relational data from various domains.

## 1 Introduction

This work studies the data that lie in the 2D/3D space and incorporate interacting relations on different spatial ranges. A representative example is the image data, where an object in the image can interact with other adjacent objects via the direct touch, and it can also interact with those distantly relevant ones via gazing, waving hands or pointing. In protein science, the protein 3D structure is another typical example, in which different amino acids can interact in short range by peptide/hydrogen bonds, and they can also interact in medium and long ranges by hydrophobic interaction. We summarize such kind of data as **spatial multi-relational data**.

In various domains, a lot of previous efforts have been made to model the spatial multi-relational data. For image modeling, multi-head self-attention mechanisms (Dosovitskiy et al., 2020; Liu et al., 2021b), convolutional operations with large receptive fields (Ding et al., 2022; Yang et al., 2022) and MLPs for mixing full spatial information (Tolstikhin et al., 2021; Touvron et al., 2021a) are explored to capture multi-range spatial interactions within an image. For protein structure modeling, Zhang et al. (2022) builds multiple groups of edges for different short-range interactions and employs relational graph convolution (Schlichtkrull et al., 2018) for multi-relational modeling. These works either implicitly treat different kinds of spatial relations (*i.e.*, short-range, medium-range and long-range relations) (Tolstikhin et al., 2021; Yang et al., 2022) or handle them by a unified scheme like relative positional encoding (Dosovitskiy et al., 2020; Liu et al., 2021b). However, considering the relative importance of these spatial relations could vary across different tasks (*e.g.*, the great importance of short-range relations in instance segmentation, and the upgraded importance of long-range relations in semantic segmentation), separately modeling each spatial relation is a better solution to capture different tasks' focus. Such a separate modeling approach remains to be explored, and, especially, the approach is expected to have efficient adaptation to large data and model scales.

To attain this goal, we propose the **EurNet** for **E**fficient m**u**lti-range **r**elational modeling. In general, EurNets are a series of relational graph neural networks equipped with graph construction layers, where relational edges are constructed by the layers for capturing multi-range spatial interactions. When instantiated with different domain knowledge (*e.g.*, computer vision or protein science), Eur-Nets can be specialized to tackle important problems like image classification, image segmentation and protein function prediction. To be specific, upon the raw data, EurNet first uses the graph construction layers to build different types of edges that respectively capture the short-, medium- and long-range spatial interactions within the data. For efficient multi-relational modeling over the constructed graph, we next introduce the *gated relational message passing* (**GRMP**) layer as the basic modeling module of EurNet. GRMP separately performs (1) relational message aggregation on each individual feature channel and (2) node-wise aggregation of different feature channels. Compared to the classical relational graph convolution (RGConv) (Schlichtkrull et al., 2018), GRMP enjoys lower computational cost when more relations are to be modeled, and thus can handle more types of spatial interactions given the same computational budget. EurNet also supports dynamic graph construction and multi-stage modeling that are used in domains like image modeling.

We demonstrate EurNets in image and protein structure modeling. To model image patches with different granularity, we build EurNets with hierarchical graph construction layers and multiple modeling stages and derive a model series with increasing capacity, *i.e.*, **EurNet-T**, **EurNet-S** and **EurNet-B**. These models enjoy comparable or better top-1 accuracy (82.3% *v.s.* 82.3%; 83.6% *v.s.* 83.5%; 84.1% *v.s.* 83.9%) against the previous SoTA FocalNet$_{\text{(LRF)}}$ series (Yang et al., 2022) on ImageNet-1K classification (resolution: $224 \times 224$). Similar performance gains are preserved on COCO object detection and ADE20K semantic segmentation. To model protein alpha carbons, we build EurNet with a single-stage model architecture as GearNet Zhang et al. (2022). Under this fix-architecture comparison, EurNet consistently outperforms the SoTA GearNet on standard protein function prediction benchmarks in terms of protein-centric maximum F-score (EC: 0.768 *v.s.* 0.730; GO-BP: 0.437 *v.s.* 0.356; GO-MF: 0.563 *v.s.* 0.503; GO-CC: 0.421 *v.s.* 0.414). These performance improvements remain when edge-level message passing is involved. Our results demonstrate that EurNet could be a strong candidate for modeling spatial multi-relational data in various domains.

## 2 RELATED WORK

**Multi-relational data modeling.** Multi-relational data are ubiquitous in the real world, *e.g.*, knowledge graphs (Toutanova & Chen, 2015) and customer-product networks (Li et al., 2014). To effectively model multiple types of relations/interactions, existing works have explored embedding-based methods (Bordes et al., 2013; Sun et al., 2019), multi-headed attention (Vaswani et al., 2017) and different relational graph neural networks (GNNs) (Schlichtkrull et al., 2018; Vashishth et al., 2019; Busbridge et al., 2019; Zhu et al., 2021). Previous relational GNNs mainly focus on model expressivity and parameter efficiency, and few works (Li et al., 2021) study the computational efficiency for relational modeling at scale. In addition, they can hardly model the spatial multi-relational data whose relational linking structures at different spatial ranges are not originally given (*e.g.*, image patches). EurNet is designed to model such kind of data in a computationally efficient way.

**Image modeling.** After the dominance of convolutional vision backbones (He et al., 2016; Tan & Le, 2019) in 2010s, researchers rethink the architectures for more effective image modeling in 2020s. Vision Transformers (Dosovitskiy et al., 2020; Liu et al., 2021b; Wang et al., 2021) replace convolutions with the self-attention mechanism (Vaswani et al., 2017) to better capture non-local interactions and gain SoTA performance. Following such successes, modern convolutional architectures (Liu et al., 2022; Yang et al., 2022), all-MLP architectures (Tolstikhin et al., 2021; Touvron et al., 2021a) and vision GNNs (Han et al., 2022) are designed to aggregate long-range spatial context. Some earlier works (Chen et al., 2019b; Zhang et al., 2019; 2020) realize non-local modeling by graph convolution on fully-connected or dynamic graphs. By comparison, EurNet captures multi-range spatial interactions from a novel graph learning perspective, *i.e.*, multi-relational modeling.

**Protein structure modeling.** A variety of protein structure encoders have been developed to acquire informative protein representations on different structural granularity, including residue-level structures (Gligorijević et al., 2021; Zhang et al., 2022), atom-level structures (Jing et al., 2021; Hermosilla et al., 2021) and protein surfaces (Gainza et al., 2020; Sverrisson et al., 2021). This work focuses on the residue-level protein structure modeling. GearNet (Zhang et al., 2022) is a closely

related work which explores multi-relational modeling of residue-level structures with short-range linking and relational graph convolution (RGConv). By comparison, our EurNet models a broader range of interactions including short, medium and long ranges, and it studies the gated relational message passing (GRMP) as a more efficient and equally effective alternative of RGConv.

## 3 EURNET FOR EFFICIENT MULTI-RANGE RELATIONAL MODELING

### 3.1 PROBLEM DEFINITION

This work studies the data $\mathcal{V} = \{v_i\}_{i=1}^N$ with $N$ data units (*e.g.*, patches in an image, alpha carbons in a protein, *etc.*) with the following structure: (1) **spatial interaction on multiple ranges**: data units can interact with each other across diverse spatial ranges; (2) **multi-relational interaction**: multiple interaction types (*i.e.*, relations) exist between different units; (3) **no canonical linking structure**: the linking structures of multi-range interactions are not specified in the raw data.

To effectively model such *spatial multi-relational data*, the model is expected to own following capabilities: (1) **dynamic multi-range linking**: the model can link relevant data units across different spatial ranges, and the linking structure can change along the whole model if desired; (2) **multi-relational linking**: the model divides all links into multiple groups based on their interaction types; (3) **efficient multi-relational modeling**: the model can propagate information among interacting units by taking their interaction types into consideration, and it will not introduce too much extra computation when involving more relations. Keeping all these requirements in mind, we next introduce the high-level designs of EurNet, and we present its detailed instantiations in Sec. 4.

### 3.2 MULTI-RANGE RELATIONAL GRAPH CONSTRUCTION

We regard each data unit $v \in \mathcal{V}$ as a node in the graph. For the lack of canonical linking structure among the nodes, we therefore seek to build edges among them, especially with considering their interactions on multiple spatial ranges and dynamically adjusting the graph structure if desired.

**Multi-range relational edge construction.** Given the concepts of spatial and semantic adjacency in a specific domain (*e.g.*, computer vision or protein science), we construct three groups of edges $\mathcal{E}_{\text{short}} = \{\{(u,v,r)\}|r \in \mathcal{R}_{\text{short}}\}$, $\mathcal{E}_{\text{medium}} = \{\{(u,v,r)\}|r \in \mathcal{R}_{\text{medium}}\}$ and $\mathcal{E}_{\text{long}} = \{\{(u,v,r)\}|r \in \mathcal{R}_{\text{long}}\}$ to represent short-, medium- and long-range spatial interactions, where $(u,v,r)$ denotes an edge from node $u$ to node $v$ with relation $r$, and $\mathcal{R}_{\text{short}}/\mathcal{R}_{\text{medium}}/\mathcal{R}_{\text{long}}$ is the set of relations for short-/medium-/long-range interactions. To capture the interactions on different spatial ranges, all these edges are gathered into the edge set $\mathcal{E} = \mathcal{E}_{\text{short}} \cup \mathcal{E}_{\text{medium}} \cup \mathcal{E}_{\text{long}} = \{\{(u,v,r)\}|r \in \mathcal{R}\}$ with the integrated relation set $\mathcal{R} = \mathcal{R}_{\text{short}} \cup \mathcal{R}_{\text{medium}} \cup \mathcal{R}_{\text{long}}$. Now, the raw data $\mathcal{V}$ is structured as a multi-relational graph $\mathcal{G} = (\mathcal{V}, \mathcal{E}, \mathcal{R})$ that is aware of diverse types of interactions within the data.

**Dynamic edge construction.** A model can focus on different levels of semantics at different modeling stages. For example, for the image modeling problem we consider, a typical hierarchical image encoder (He et al., 2016; Liu et al., 2021b) is split into multiple stages, and it tends to encode low-level features in shallower stages and encode high-level semantics in deeper stages. To accommodate such a hierarchical modeling manner, our graph construction scheme will be dynamically performed before each modeling stage based on the input features (*e.g.*, node coordinates or representations) of the stage, so that each modeling stage can explore its specific neighborhood structures of data units.

### 3.3 GATED RELATIONAL MESSAGE PASSING

To perform multi-relational modeling over the constructed graph $\mathcal{G}$, the typical method Relational Graph Convolution (RGConv) (Schlichtkrull et al., 2018) employs a unique convolutional kernel matrix $W_r$ to aggregate the messages of relation $r$, leading to $|\mathcal{R}|$ different kernel matrices in total for the message aggregation from neighborhoods. Taking node $v$ as an example, the RGConv layer updates its representation from $z_v$ to $z_v'$ as below:

$$z_v^{\text{aggr}} = \sum_{r \in \mathcal{R}} \sum_{u \in \mathcal{N}_r(v)} \frac{1}{|\mathcal{N}_r(v)|} W_r z_u, \quad z_v' = W^{\text{self}} z_v + z_v^{\text{aggr}}, \tag{1}$$

where $z_v^{\text{aggr}}$ is the aggregated message for node $v$, $\mathcal{N}_r(v) = \{u|(u,v,r) \in \mathcal{E}\}$ are $v$'s neighbors with relation $r$, and $W^{\text{self}}$ is the weight matrix for self-update (we omit all bias terms for brevity).

We assume that, when introducing a new relation, the in-degree of each node will increase by $\bar{d}$ on average. By taking the efficient implementation of RGConv with sparse matrix multiplication, it can be shown that the floating-point operations (FLOPs) of RGConv with $C$-dimensional input and output node features has the following form (see Appendix A for proof):

$$\text{FLOPs(RGConv)} = |\mathcal{R}| \cdot (2\bar{d}|\mathcal{V}|C + 2|\mathcal{V}|C^2) + 2|\mathcal{V}|C^2 + |\mathcal{V}|C. \tag{2}$$

Therefore, the computational cost will scale with the relation number $|\mathcal{R}|$ by the factor of $2\bar{d}|\mathcal{V}|C + 2|\mathcal{V}|C^2$. Considering both the node number $|\mathcal{V}|$ and the feature dimension $C$ could be large in many applications, the $2|\mathcal{V}|C^2$ term will be the main obstacle of exploring more relations with moderate extra computation, which hurts the model capacity under a strict constraint on computational cost.

For more efficient multi-relational modeling, we aim at an approach that (1) can effectively model the interactions among relational messages and among feature channels, and (2) owns a gentle scaling behavior when modeling increasing number of relations within the data. To attain this goal, we propose the Gated Relational Message Passing (GRMP). Inspired by light-weight separable graph convolution methods (Balcilar et al., 2020; Li et al., 2021) that aggregate neighborhood features in a channel-wise way, GRMP decomposes the relation-channel entangled aggregation of RGConv into (*i*) the aggregation of intra- and inter-relation messages on each individual channel and (*ii*) the aggregation of different feature channels. Specifically, it consecutively performs following steps: ① a pre-layer node-wise channel aggregation with the weight matrix $W^{\text{in}}$, ② an intra-relation message aggregation through channel-wise graph convolution, ③ an inter-relation message aggregation by node-adaptive weighted summation, ④ a post-layer node-wise channel aggregation with the weight matrix $W^{\text{out}}$, and ⑤ the final node representation update by *regarding the aggregated neighborhood information as gate*. Formally, GRMP updates the representation of node $v$ from $z_v$ to $z_v'$ as below:

$$z_v^{\text{aggr}} = W^{\text{out}} \overbrace{\left( \sum_{r \in \mathcal{R}} \alpha_r(v) \cdot \underbrace{\sum_{u \in \mathcal{N}_r(v)} \frac{1}{|\mathcal{N}_r(v)|}}_{\text{step ②}} w_r \odot \underbrace{(W^{\text{in}} z_u)}_{\text{step ①}} \right)}^{\substack{\text{step ④} \\ \text{step ③}}}, \quad z_v' = \underbrace{W^{\text{self}} z_v \odot z_v^{\text{aggr}}}_{\text{step ⑤}}, \tag{3}$$

where $\alpha(v) = W^{\alpha} z_v \in \mathbb{R}^{|\mathcal{R}|}$ are the attentive weights assigned to all relations on node $v$ ($W^{\alpha}$ is the weight matrix for node-adaptive relation weighting), $w_r$ is the channel-wise convolutional kernel vector for relation $r$ (with the same shape as the node feature vector after step ①), and $\odot$ denotes the Hadamard product. The definitions of $z_v^{\text{aggr}}$, $\mathcal{N}_r(v)$ and $W^{\text{self}}$ follow Eq. (1), and all biases are omitted. We analyze the components of GRMP in Appendix H.1 and its expressivity in Appendix B. We also provide a graphical illustration of the GRMP layer in Appendix C.

Under the efficient implementation with sparse matrix multiplication, GRMP consumes the FLOPs as below when taking $C$-dimensional input and output node features (see Appendix A for proof):

$$\text{FLOPs(GRMP)} = |\mathcal{R}| \cdot (2\bar{d} + 7)|\mathcal{V}|C + 6|\mathcal{V}|C^2. \tag{4}$$

Therefore, the relation number $|\mathcal{R}|$ scales the computational cost of GRMP with the scaling factor $(2\bar{d} + 7)|\mathcal{V}|C$. Compared to the scaling factor $2\bar{d}|\mathcal{V}|C + 2|\mathcal{V}|C^2$ of RGConv, this factor gets rid of the quadratic reliance on feature dimension and thus leads to a gentler scaling behavior when increasing the number of considered relations. In Fig. 1, we compare the FLOPs of RGConv and GRMP when they respectively serve as the building block of EurNet-T for image modeling (image resolution: $224 \times 224$; "T" denotes the tiny-scale model). In this illustrative comparison, we simply connect each node (*i.e.*, image patch) with its $K$-nearest neighbors in terms of representation similarity, and the connection with the $k$-th nearest neighbor is regarded as the $k$-th relation, leading to $K$ relations in total. We can observe that, when increasing

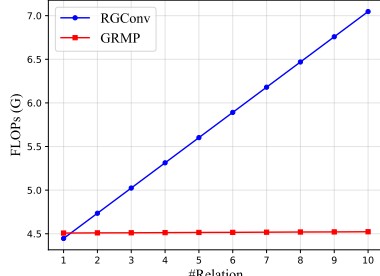

Figure 1: FLOPs trend of RGConv and GRMP under different relation numbers, evaluated on EurNet-T for image modeling.

the number of neighbors and thus the number of relations, the computational cost of GRMP-based model increases much more gently than the RGConv-based one. This merit enhances the efficiency and effectiveness of GRMP-based models in real-world problems like image and protein structure modeling, as studied in the second paragraph of Sec. 5.3 and in the Appendix H.3.

## 4 INSTANTIATIONS OF EURNET

In the main paper, we focus on two application domains, *i.e.*, computer vision and protein science, where modeling spatial multi-relational data (*i.e.*, images and protein structures) can solve important problems. In Appendix G, we further study the effectiveness of EurNet on modeling an important kind of multi-relational data without spatial information, *i.e.*, knowledge graphs.

### 4.1 EURNET FOR IMAGE MODELING

#### 4.1.1 RELATIONAL EDGES FOR SHORT-, MEDIUM- AND LONG-RANGE INTERACTIONS

Following previous practices (Dosovitskiy et al., 2020; Liu et al., 2021b), we split an image into local patches and regard these patches as the node set $\mathcal{V}$ of our multi-relational graph. Upon these patches, we construct following relational edges to capture different ranges of spatial interactions within an image (see Fig. 2 for a graphical illustration):

- **Edges for short-range interactions** ($|\mathcal{R}_{\text{short}}| = 4$). We connect each patch with its up, down, left and right patches and regard each direction of adjacency as a relation. These edges capture the one-hop spatial neighbors and thus shortest-range spatial interactions of each image patch.

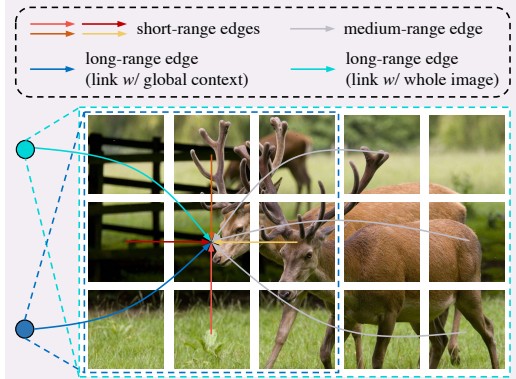

- **Edges for medium-range interactions** ($|\mathcal{R}_{\text{medium}}| = 1$). In the medium range, a patch can interact with other patches sharing similar semantics (*e.g.*, different body parts of deer in Fig. 2). We thus connect each patch with its $K$-nearest neighbors in terms of representation similarity measured by negative Euclidean distance (we analyze

Figure 2: Multi-range relational edges for image.

the sensitivity of $K$ in Appendix I), and these edges are with the same relation. All edges connecting two patches within the same 2×2 window are removed to avoid short-range linking.

- **Edges for long-range interactions** ($|\mathcal{R}_{\text{long}}| = 2$). To model long-range interactions, we introduce two kinds of *virtual nodes* and the associated edges. (1) *A virtual node for whole-image representation* is derived by global average pooling over all patch representations, and this virtual node is linked to all patches. (2) *Per-patch virtual nodes for surrounding global context* are got by a stack of depth-wise 2D convolutions (Yang et al., 2022) that aggregate each patch's contextual information with large receptive field and low computation; an edge links each of these virtual nodes to its corresponding patch with a different long-range interacting relation against that in (1), due to the different global context levels represented by two kinds of virtual nodes.

By gathering all these edges representing 7 different relations, we have the edge set $\mathcal{E}$, the relation (*i.e.*, edge type) set $\mathcal{R}$ and the full graph $\mathcal{G} = (\mathcal{V}, \mathcal{E}, \mathcal{R})$ for multi-relational image modeling.

#### 4.1.2 MODEL ARCHITECTURE

**General architecture.** In general, we follow the hierarchical image modeling architecture proposed by Swin Transformer (Liu et al., 2021b), which is verified to be a superior architecture and is applied to many vision backbones (Liu et al., 2022; Yang et al., 2021; 2022). Specifically, the whole model is divided into four stages that (1) reduce the number of patches (*i.e.*, nodes in our graph) to a quarter across consecutive stages, and (2) use increasing number of feature channels $[C, 2C, 4C, 8C]$ for all stages. Each stage contains multiple modeling blocks, where we construct each block with a GRMP layer (Sec. 3.3) for relational message passing and a feed-forward network (FFN) (Vaswani et al., 2017) for feature transformation. We adjust the number of feature channels and the number of blocks in each stage to get a model series with increasing capacity, *i.e.*, **EurNet-T**, **EurNet-S** and **EurNet-B**. The detailed architectures of these models are displayed in Appendix D.

**Graph construction layers.** To adapt the multi-stage modeling manner, we put a graph construction layer before each modeling stage of EurNet-T/S/B. In this way, based on the locations and representations of the patches fed into each stage, the multi-relational graph $\mathcal{G}$ will be reconstructed to adapt

these stage inputs. In particular, along the modeling stages, the edges for medium-range interactions are expected to capture the semantic neighbors on different semantic levels (*e.g.*, from the relevance of low-level features to the relevance of high-level semantics), as studied in Sec. 5.4.

## 4.2 EURNET FOR PROTEIN STRUCTURE MODELING

### 4.2.1 RELATIONAL EDGES FOR SHORT-, MEDIUM- AND LONG-RANGE INTERACTIONS

In this work, we consider the alpha carbon (*i.e.*, $C\alpha$) graph as the representation of protein structure, which is an informative and light-weight summary of the overall protein 3D structure and is widely used in the literature (Gligorijević et al., 2021; Baldassarre et al., 2021; Zhang et al., 2022) (see Appendix E for a preliminary introduction to protein structure). In specific, we extract all $C\alpha$s as the node set $\mathcal{V}$ of our graph, which, at this time, is actually a set of separate points in the 3D space, since there is no chemical bond among $C\alpha$s. To describe the multi-range spatial interactions within a protein, we build following relational edges (see Fig. 3 for a graphical illustration):

- **Edges for short-range interactions** ($|\mathcal{R}_{\text{short}}| = 6$). We adopt two kinds of short-range edges proposed by Zhang et al. (2022). (1) *Sequential edges* connect the $C\alpha$ nodes that are within the distance of 2 on the protein sequence, where each of the sequential distances $\{-2,-1,0,1,2\}$ is regarded as a single relation (*i.e.*, 5 relations in total). (2) *Radius edges* connect the $C\alpha$ nodes within the Euclidean distance of 10 angstroms, and all radius edges have the same relation.

- **Edges for medium-range interactions** ($|\mathcal{R}_{\text{medium}}| = 2$). To capture medium-range interactions exclusively, for each $C\alpha$ node, we first filter out all its neighbors within the sequential distance of 5 or within the Euclidean distance of 10 angstroms. We then connect it with the remaining nodes that are *5 nearest* and *5~10 nearest* to it (measured by Euclidean distance), and the connections with these two sets of neighbors are regarded as two different relations.

- **Edges for long-range interactions** ($|\mathcal{R}_{\text{long}}| = 1$). To capture the interactions beyond short- and medium-range interactions above, we introduce a *virtual node* representing the whole protein by taking global average pooling over all $C\alpha$ representations, and this virtual node is linked to all $C\alpha$ nodes with a single relation. These edges make each $C\alpha$ aware of the status of all other $C\alpha$s, and thus the long-range interactions beyond short and medium ranges can be captured.

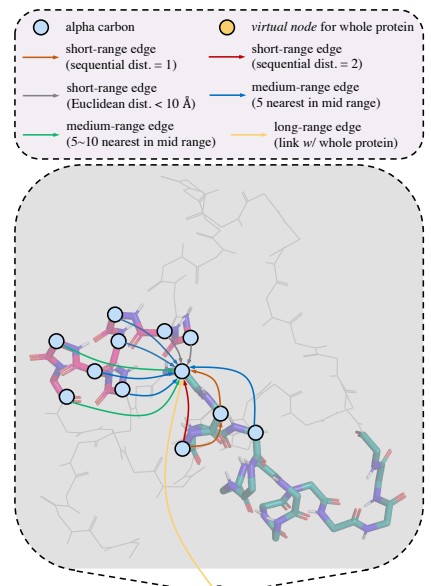

Figure 3: Multi-range relational edges for protein $C\alpha$s. *Abbr.*, dist.: distance.

We gather all these edges with 9 different relations into the edge set $\mathcal{E}$ and the relation set $\mathcal{R}$, which, together with $\mathcal{V}$, derive the full graph $\mathcal{G} = (\mathcal{V}, \mathcal{E}, \mathcal{R})$ for multi-relational protein structure modeling.

### 4.2.2 MODEL ARCHITECTURE

This work focuses on comparing the graph construction and message passing schemes of EurNet against the SoTA GearNet (Zhang et al., 2022), and we thus follow its single-stage model architecture for fair comparison. Specifically, EurNet performs graph construction once before this only modeling stage, and the input node feature is the one-hot encoding of each $C\alpha$'s corresponding amino acid. Upon these inputs, six GRMP layers (Sec. 3.3) are stacked for relational modeling. After each layer, the sum pooling over all $C\alpha$ representations is deemed as the whole-protein representation, and these per-layer protein representations are concatenated to produce the final output. Upon this output, EurNet performs a downstream task by appending a task-specific prediction head. We leave the design of the protein structure encoder with multiple modeling stages as a future work.

*Note that*, in EurNet, all graph construction and message passing operations rely only on the quantities (*e.g.*, sequential and Euclidean distance) that are invariant to translation, rotation and reflection. Therefore, EurNet satisfies **E(3)-invariance** (Mumford et al., 1994).

## 5 EXPERIMENTS

### 5.1 PERFORMANCE COMPARISON ON IMAGE MODELING

#### 5.1.1 BASELINE METHODS

We do point-by-point comparisons between our EurNet series, the SoTA ConvNeXt (Liu et al., 2022) and FocalNet (Yang et al., 2022) series, and other standard series including Swin Transformer (Liu et al., 2021b), FocalAtt (Yang et al., 2021) and ViG (Han et al., 2022). For completeness, we also report the results of EffNet (Tan & Le, 2019), EffNetV2 (Tan & Le, 2021), ViT (Dosovitskiy et al., 2020), DeiT (Touvron et al., 2021b), PVT (Wang et al., 2021), Mixer (Tolstikhin et al., 2021), gMLP (Liu et al., 2021a) and ResMLP (Touvron et al., 2021a) in applicable cases.

#### 5.1.2 IMAGE CLASSIFICATION ON IMAGENET-1K

**Setups.** In this set of experiments, we benchmark the classification performance of different backbones on ImageNet-1K (Deng et al., 2009) (with 1.28M training and 50K validation images from 1,000 classes) in terms of top-1 accuracy. We consider both (1) training on ImageNet-1K from scratch and (2) pretraining on ImageNet-22K followed by ImageNet-1K fine-tuning. For fair comparison, we follow the standard training configurations of Swin Transformer (Liu et al., 2021b) with minor changes. Detailed model and training configurations are stated in Appendix F.1.

**Results.** In Tab. 1, for training on ImageNet-1K from scratch, EurNets outperform or align previous SoTA baselines on $224^2$ image size, *i.e.*, EurNet-T *v.s.* FocalNet$_{(LRF)}$-T: **82.3**% *v.s.* **82.3**%; EurNet-S *v.s.* FocalNet$_{(LRF)}$-S: **83.6**% *v.s.* 83.5%; EurNet-B *v.s.* FocalNet$_{(LRF)}$-B: **84.1**% *v.s.* 83.9%. Following Swin Transformer, we lift the resolution to $384^2$ for 30 epochs fine-tuning after training EurNet-B for 300 epochs on the $224^2$ resolution, this model gains **85.4**% top-1 accuracy, outperforming ConvNeXt-B. These results demonstrate the effectiveness of EurNets on modeling images with different resolutions.

For ImageNet-22K pre-training, EurNet-B is on par with the previous SoTA ConvNeXt-B and clearly outperforms Swin-B. The pre-training on large scale is widely regarded as the strength of the models with few inductive biases like Swin-B; while our results show that the well-designed EurNet-B with more inductive biases could also be effective, aligning with ConvNeXt's finding.

**Throughput analysis.** The throughput of EurNet is higher than FocalAtt while lower than FocalNet and ConvNeXt. We point out that 2D convolu-

Table 1: ImageNet-1K classification results. We measure throughput on a V100 GPU. † denotes the model pre-trained on ImageNet-22K. $224^2$ and $384^2$ denote the image size. "↑384" means fine-tuning on 384×384 images for 30 epochs.

| Model | #Params. (M) | FLOPs (G) | Throughput (imgs/s) | Top-1 Acc (%) |
|---|---|---|---|---|
| **Train on ImageNet-1K from scratch** | | | | |
| EffNet-B7 | 66 | 37.0 | 55 | 84.3 |
| EffNetV2-L | 120 | 53.0 | 84 | 85.7 |
| ViT-S/16 | 22 | 4.6 | 939 | 79.9 |
| ViT-B/16 | 87 | 17.6 | 330 | 81.8 |
| DeiT-S/16 | 22 | 4.6 | 979 | 79.8 |
| DeiT-S/16 | 87 | 17.6 | 302 | 81.8 |
| PVT-Small | 25 | 3.8 | 794 | 79.8 |
| PVT-Medium | 44 | 6.7 | 517 | 81.2 |
| PVT-Large | 61 | 9.8 | 352 | 81.7 |
| Mixer-B/16 | 60 | 12.7 | 455 | 76.4 |
| gMLP-S | 20 | 4.5 | 785 | 79.6 |
| gMLP-B | 73 | 15.8 | 301 | 81.6 |
| ResMLP-S24 | 30 | 6.0 | 871 | 79.4 |
| ResMLP-B24 | 129 | 23.0 | 61 | 81.0 |
| Swin-T | 28 | 4.9 | 760 | 81.2 |
| Pyramid ViG-S | 27 | 4.6 | – | 82.1 |
| FocalAtt-T | 29 | 4.9 | 319 | 82.2 |
| FocalNet$_{(LRF)}$-T | 29 | 4.5 | 696 | **82.3** |
| ConvNeXt-T | 29 | 4.5 | 775 | 82.1 |
| EurNet-T | 29 | 4.6 | 530 | **82.3** |
| Swin-S | 50 | 8.7 | 435 | 83.1 |
| Pyramid ViG-M | 52 | 8.9 | – | 83.1 |
| FocalAtt-S | 51 | 9.4 | 192 | 83.5 |
| FocalNet$_{(LRF)}$-S | 50 | 8.7 | 406 | 83.5 |
| ConvNeXt-S | 50 | 8.7 | 447 | 83.1 |
| EurNet-S | 50 | 8.8 | 314 | **83.6** |
| Swin-B | 88 | 15.4 | 291 | 83.5 |
| Pyramid ViG-B | 93 | 16.8 | – | 83.7 |
| FocalAtt-B | 90 | 16.4 | 138 | 83.8 |
| FocalNet$_{(LRF)}$-B | 89 | 15.4 | 269 | 83.9 |
| ConvNeXt-B | 89 | 15.4 | 292 | 83.8 |
| EurNet-B | 89 | 15.6 | 224 | **84.1** |
| Swin-B↑384 | 88 | 47.1 | 85 | 84.5 |
| ConvNeXt-B↑384 | 89 | 45.0 | 96 | 85.1 |
| EurNet-B↑384 | 90 | 46.6 | 69 | **85.4** |
| **Pre-train on ImageNet-22K & Fine-tune on ImageNet-1K** | | | | |
| Swin-B† ($224^2$) | 88 | 15.4 | 291 | 85.2 |
| FocalNet$_{(SRF)}$-B† ($224^2$) | 88 | 15.3 | 280 | 85.6 |
| ConvNeXt-B† ($224^2$) | 89 | 15.4 | 292 | **85.8** |
| EurNet-B† ($224^2$) | 89 | 15.6 | 224 | 85.7 |
| Swin-B† ($384^2$) | 88 | 47.0 | 85 | 86.4 |
| FocalNet$_{(SRF)}$-B† ($384^2$) | 88 | 44.8 | 94 | 86.5 |
| ConvNeXt-B† ($384^2$) | 89 | 45.1 | 96 | 86.8 |
| EurNet-B† ($384^2$) | 90 | 46.6 | 69 | **87.0** |

tions (*i.e.*, the core of FocalNet and ConvNeXt) are well supported by CUDA kernels, while such supports are still ongoing for graph operations (Chen et al., 2020; Min et al., 2021). EurNet's further speedup is expected under maturer CUDA supports.

#### 5.1.3 OBJECT DETECTION ON COCO

**Setups.** This experiment benchmarks the object detection and instance segmentation performance on COCO 2017 (Lin et al., 2014). All models are trained on 118K training images and evaluated on

Table 2: COCO object detection and instance segmentation results with Mask R-CNN (He et al., 2017).

| Model | #Params. (M) | FLOPs (G) | Mask R-CNN 1× | | | | | | Mask R-CNN 3× | | | | | |
|---|---|---|---|---|---|---|---|---|---|---|---|---|---|---|
| | | | $AP^b$ | $AP^b_{50}$ | $AP^b_{75}$ | $AP^m$ | $AP^m_{50}$ | $AP^m_{75}$ | $AP^b$ | $AP^b_{50}$ | $AP^b_{75}$ | $AP^m$ | $AP^m_{50}$ | $AP^m_{75}$ |
| PVT-Small | 44.1 | 245 | 40.4 | 62.9 | 43.8 | 37.8 | 60.1 | 40.3 | 43.0 | 65.3 | 46.9 | 39.9 | 62.5 | 42.8 |
| Swin-T | 47.8 | 264 | 43.7 | 66.6 | 47.7 | 39.8 | 63.3 | 42.7 | 46.0 | 68.1 | 50.3 | 41.6 | 65.1 | 44.9 |
| FocalAtt-T | 48.8 | 291 | 44.8 | 67.7 | 49.2 | 41.0 | 64.7 | 44.2 | 47.2 | 69.4 | 51.9 | 42.7 | **66.5** | 45.9 |
| FocalNet$_{(LRF)}$-T | 48.9 | 268 | **46.1** | 68.2 | **50.6** | 41.5 | 65.1 | 44.5 | **48.0** | **69.7** | **53.0** | **42.9** | **66.5** | **46.1** |
| EurNet-T | 49.8 | 271 | **46.1** | **68.7** | 50.5 | **41.6** | **65.5** | **44.6** | 47.8 | 69.5 | 52.3 | **42.9** | **66.5** | **46.1** |
| PVT-Medium | 63.9 | 302 | 42.0 | 64.4 | 45.6 | 39.0 | 61.6 | 42.1 | 44.2 | 66.0 | 48.2 | 40.5 | 63.1 | 43.5 |
| Swin-S | 69.1 | 354 | 46.5 | 68.7 | 51.3 | 42.1 | 65.8 | 45.2 | 48.5 | 70.2 | 53.5 | 43.3 | 67.3 | 46.6 |
| FocalAtt-S | 71.2 | 401 | 47.4 | 69.8 | 51.9 | 42.8 | 66.6 | 46.1 | 48.8 | 70.5 | 53.6 | 43.8 | 67.7 | 47.2 |
| FocalNet$_{(LRF)}$-S | 72.3 | 365 | 48.3 | **70.5** | 53.1 | 43.1 | **67.4** | **46.2** | 49.3 | **70.7** | 54.2 | 43.8 | **67.9** | 47.4 |
| EurNet-S | 72.8 | 364 | **48.4** | **70.5** | **53.2** | **43.2** | **67.4** | **46.2** | **49.4** | **70.7** | **54.5** | **44.0** | 67.6 | **47.5** |
| PVT-Large | 81.0 | 364 | 42.9 | 65.0 | 46.6 | 39.5 | 61.9 | 42.5 | 44.5 | 66.0 | 48.3 | 40.7 | 63.4 | 43.7 |
| Swin-B | 107.1 | 497 | 46.9 | 69.2 | 51.6 | 42.3 | 66.0 | 45.5 | 48.5 | 69.8 | 53.2 | 43.4 | 66.8 | 46.9 |
| FocalAtt-B | 110.0 | 533 | 47.8 | 70.2 | 52.5 | 43.2 | 67.3 | 46.5 | 49.0 | 70.1 | 53.6 | 43.7 | 67.6 | 47.0 |
| FocalNet$_{(LRF)}$-B | 111.4 | 507 | 49.0 | 70.9 | 53.9 | 43.5 | 67.9 | 46.7 | 49.8 | 70.9 | 54.6 | 44.1 | 68.2 | 47.2 |
| EurNet-B | 112.1 | 506 | **49.3** | **71.8** | **54.0** | **43.9** | **68.2** | **47.2** | **50.1** | **71.5** | **55.1** | **44.5** | **68.7** | **47.8** |

5K validation images. Two standard training schedules, *i.e.*, the 1× schedule with 12 epochs and the 3× schedule with 36 epochs, are used for benchmarking. Detailed setups are stated in Appendix F.2.

**Results.** In Tab. 2, EurNet performs comparably to FocalNet$_{(LRF)}$ on the tiny and small model scales. We can observe the superiority of EurNet-B over FocalNet$_{(LRF)}$-B on the base model scale (better performance on all 12 metrics). The base-scale EurNet-B owns $[2, 2, 18, 2]$ modeling blocks (more than EurNet-T) and $[128, 256, 512, 1024]$ feature channels (more than EurNet-S) for four modeling stages. Therefore, *larger message passing hops* (achieved by more modeling blocks) coupled with larger model width favor EurNet's performance on high-resolution dense prediction tasks.

### 5.1.4 SEMANTIC SEGMENTATION ON ADE20K

**Setups.** In this experiment, we benchmark the semantic segmentation performance of different backbones on ADE20K (Zhou et al., 2017) which contains 20K training, 2K validation and 3K test images. The mIoU metrics under both single- and multi-scale (MS) evaluation are reported. We provide more details in Appendix F.3.

**Results.** Tab. 3 reports all results. It can be observed that EurNet-T, EurNet-S and EurNet-B achieve the best performance on their corresponding model scales under both evaluation metrics. Such consistent performance gains verify the effectiveness of EurNet on the dense prediction tasks that require to model fine-grained semantics and long-range interactions.

Table 3: ADE20K semantic segmentation results with UperNet (Xiao et al., 2018).

| Model | #Params. (M) | FLOPs (G) | mIoU | +MS |
|---|---|---|---|---|
| Swin-T | 60 | 941 | 44.5 | 45.8 |
| FocalAtt-T | 62 | 998 | 45.8 | 47.0 |
| ConvNeXt-T | 60 | 939 | - | 46.7 |
| FocalNet$_{(LRF)}$-T | 61 | 949 | 46.8 | 47.8 |
| EurNet-T | 62 | 948 | **47.2** | **48.4** |
| Swin-S | 81 | 1038 | 47.6 | 49.5 |
| FocalAtt-S | 85 | 1130 | 48.0 | 50.0 |
| ConvNeXt-S | 82 | 1027 | - | 49.6 |
| FocalNet$_{(LRF)}$-S | 84 | 1044 | 49.1 | 50.1 |
| EurNet-S | 85 | 1042 | **49.8** | **50.8** |
| Swin-B | 121 | 1188 | 48.1 | 49.7 |
| FocalAtt-B | 126 | 1354 | 49.0 | 50.5 |
| ConvNeXt-B | 122 | 1170 | - | 49.9 |
| FocalNet$_{(LRF)}$-B | 126 | 1192 | 50.5 | 51.4 |
| EurNet-B | 126 | 1190 | **50.7** | **51.8** |

## 5.2 PERFORMANCE COMPARISON ON PROTEIN STRUCTURE MODELING

### 5.2.1 BASELINE METHODS

We compare with the SoTA GearNet (Zhang et al., 2017) under two settings, *i.e.*, with and without edge message passing ("-Edge" in Tab. 4). We also include other baselines, *i.e.*, 3DCNN_MQA (Derevyanko et al., 2018), GCN (Kipf & Welling, 2016), GAT (Veličković et al., 2017), GVP (Jing et al., 2021), GraphQA (Baldassarre et al., 2021) and New IEConv (Hermosilla & Ropinski, 2022), for complete comparisons.

### 5.2.2 PROTEIN FUNCTION PREDICTION

Table 4: $F_{max}$ results on EC and GO protein function prediction benchmarks.

| Model | EC | GO-BP | GO-MF | GO-CC |
|---|---|---|---|---|
| 3DCNN_MQA | 0.077 | 0.240 | 0.147 | 0.305 |
| GCN | 0.320 | 0.252 | 0.195 | 0.329 |
| GAT | 0.368 | 0.284 | 0.317 | 0.385 |
| GVP | 0.489 | 0.326 | 0.426 | 0.420 |
| GraphQA | 0.509 | 0.308 | 0.329 | 0.413 |
| New IEConv | 0.735 | 0.374 | 0.544 | **0.444** |
| GearNet | 0.730 | 0.356 | 0.503 | 0.414 |
| EurNet | **0.768** | **0.437** | **0.563** | **0.421** |
| GearNet-Edge | 0.810 | 0.403 | 0.580 | 0.450 |
| EurNet-Edge | **0.829** | **0.456** | **0.592** | **0.453** |

**Setups.** This set of experiments compare different protein structure encoders on the EC (Gligorijević et al., 2021) and GO (Gligorijević et al., 2021) protein function prediction benchmarks. We follow GearNet to report the protein-centric maximum F-score $F_{max}$, a commonly-used metric in CAFA challenges (Radivojac et al., 2013). More dataset, model and training details are in Appendix F.4.

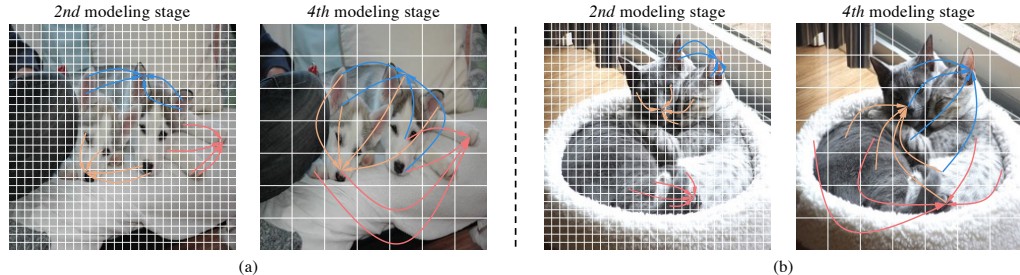

|  2nd modeling stage | 4th modeling stage | 2nd modeling stage | 4th modeling stage |

(a)      (b)

Figure 4: Medium-range edges built by EurNet-T (we use different colors for different selected target nodes).

**Results.** In Tab. 4, we can observe that EurNet consistently outperforms GearNet on all four tasks, and the performance gains preserve after involving edge message passing (details of edge message passing are stated in Appendix F.4). Since EurNet follows the single-stage model architecture of GearNet, we can conclude the effectiveness of *medium- and long-range interaction modeling* and *GRMP-based multi-relational modeling*, which are novel modeling mechanisms in EurNet.

### 5.3 ABLATION STUDY

**Effect of multi-range relational edges.** In Tab. 5, we evaluate EurNet-T on ImageNet-1K with different ranges of edges. When using a single range, the model with long-range edges achieves the highest accuracy 81.7%, which verifies the importance of capturing long-range interactions in image classification. By further adding short- or medium-range edges, the performance is promoted to 82.0%, where more fine-grained local interactions are captured. By using all three ranges of

Table 5: Ablation study of multi-range edges on ImageNet-1K with EurNet-T.

| short | medium | long | Top-1 Acc (%) |
|:---:|:---:|:---:|:---:|
| ✓ | | | 80.7 |
| | ✓ | | 79.3 |
| | | ✓ | **81.7** |
| ✓ | ✓ | | 81.5 |
| ✓ | | ✓ | **82.0** |
| | ✓ | ✓ | **82.0** |
| ✓ | ✓ | ✓ | **82.3** |

edges, the full model of EurNet-T obtains the 82.3% accuracy, which proves the complementarity of short-, medium- and long-range edges. Ablation study for protein structure is in Appendix H.2.

**Effect of GRMP layer.** In Tab. 6, we compare between RGConv and GRMP under the comparable parameter number, FLOPs and throughput. (1) GRMP's dimensions are first set as $[96, 192, 384, 768]$ in four stages. To reach comparable cost, RGConv can only have the dimensions of

Table 6: Ablation study of multi-relational modeling layer on ImageNet-1K with EurNet-T.

| Layer | Hidden Dimensions | #Params. (M) | FLOPs (G) | Throughput (imgs/s) | Top-1 Acc (%) |
|:---|:---:|:---:|:---:|:---:|:---:|
| RGConv | $[84, 168, 336, 672]$ | 28.8 | 4.6 | 541.8 | 81.5 |
| GRMP | $[96, 192, 384, 768]$ | 28.8 | 4.6 | 530.3 | **82.3** |
| RGConv | $[96, 192, 384, 768]$ | 37.3 | 5.9 | 451.2 | 82.2 |
| GRMP | $[108, 216, 432, 864]$ | 36.3 | 5.8 | 444.5 | **82.7** |

$[84, 168, 336, 672]$ and achieves a lower accuracy 81.5% than GRMP's 82.3%. (2) After increasing RGConv's dimensions to $[96, 192, 384, 768]$, it aligns GRMP's performance while introduces more cost (1.3G more FLOPs). Under comparable cost, GRMP can have $[108, 216, 432, 864]$ dimensions, leading to a higher accuracy 82.7%. These results demonstrate the better efficiency-performance trade-off gained by GRMP. Ablation study for protein structure modeling is in Appendix H.3.

### 5.4 VISUALIZATION

Fig. 4 displays some medium-range edges built by the EurNet-T trained on ImageNet-1K. The edges for the 2nd stage connect the patches with similar low-level features (*e.g.*, the patches of red dog ears in Fig. 4(a)), while the edges for the 4th stage connect semantically relevant patches (*e.g.*, different body parts of two dogs in Fig. 4(a)), which shows EurNet-T's hierarchical image modeling ability.

## 6 CONCLUSIONS AND FUTURE WORK

This work proposes the EurNet to model spatial multi-relational data like image patches and protein alpha carbons. It builds relational edges on multiple spatial ranges to describe the interactions in the data. It uses the gated relational message passing layer to model the built multi-relational graph, which can efficiently adapt to large data and model scales. The instantiations of EurNet have gained superior performance on various image and protein structure modeling tasks.

In future works, we will adapt EurNet to more tasks of other domains like 3D point cloud modeling for object and scene understanding, and we will explore a general hierarchical multi-relational modeling method for the data from various domains.

## REPRODUCIBILITY STATEMENT

For the sake of reproducibility, we use Tab. 7 to provide detailed architectures of EurNet-T, EurNet-S and EurNet-B for image modeling, state the detailed single-stage architecture of EurNet for protein structure modeling in Sec. 4.2.2, and describe the model configurations for each specific task in Sec. F. We state the detailed training configurations of all considered tasks in Sec. F. The derivation of the FLOPs of RGConv (Eq. (2)) and GRMP (Eq. (4)) are provided in Sec. A. In the supplementary material, we submit all source code for reproducing the results of ImageNet classification experiments, and the source code for COCO object detection, ADE20K semantic segmentation and EC and GO protein function prediction experiments will be released to public upon acceptance.

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

## A   FLOPS OF RGCONV AND GRMP

For FLOPs computation, we consider the multi-relational graph $\mathcal{G} = (\mathcal{V}, \mathcal{E}, \mathcal{R})$ with node set $\mathcal{V}$, edge set $\mathcal{E}$ and relation (*i.e.*, edge type) set $\mathcal{R}$, and both input and output node features are with $C$ feature channels. In addition, we assume that, when introducing a new relation, the in-degree of each node will increase by $\bar{d}$ on average.

**Proposition 1.** *To process the assumed multi-relational graph, the Relational Graph Convolution (RGConv) consumes the FLOPs as below under the efficient implementation with sparse matrix multiplication:*

$$\text{FLOPs}(\text{RGConv}) = |\mathcal{R}| \cdot (2\bar{d}|\mathcal{V}|C + 2|\mathcal{V}|C^2) + 2|\mathcal{V}|C^2 + |\mathcal{V}|C.$$

*Proof.* We divide the computation of RGConv into three steps and compute the FLOPs of each step:

① In the first step, the adjacency of all node pairs on $|\mathcal{R}|$ different relations are summarized in the adjacency matrix $A \in \mathbb{R}^{|\mathcal{V}| \times |\mathcal{R}||\mathcal{V}|}$, where the element $A_{i,(j-1)|\mathcal{R}|+k}$ indicates the weight of the edge from the $i$-th node to the $j$-th node with the $k$-th relation:

$$A_{i,(j-1)|\mathcal{R}|+k} = \begin{cases} \frac{1}{|\mathcal{N}_{r_k}(v_j)|} & \text{there is an edge from } i\text{-th node to } j\text{-th node with } k\text{-th relation,} \\ 0 & \text{otherwise,} \end{cases} \tag{5}$$

where $\mathcal{N}_{r_k}(v_j) = \{u|(u, v_j, r_k) \in \mathcal{E}\}$ is the neighborhood set of node $v_j$ with relation $r_k$. Using this adjacency matrix, each node will have $|\mathcal{R}|$ different slots to receive the relational messages passed to it. All relational message passing operations can be realized by a sparse matrix multiplication:

$$\tilde{Z} = A^\top Z, \tag{6}$$

where $Z \in \mathbb{R}^{|\mathcal{V}| \times C}$ denotes input node features, and $\tilde{Z} \in \mathbb{R}^{|\mathcal{R}||\mathcal{V}| \times C}$ denotes the relational slots of all nodes after message passing. By utilizing the sparsity of the adjacency matrix, this step consumes following FLOPs:

$$\text{FLOPs}(\text{RGConv}-①) = 2|\mathcal{E}|C = 2\bar{d}|\mathcal{R}||\mathcal{V}|C. \tag{7}$$

② In the second step, we first integrate the relational slots of each node to get the reshaped $\tilde{Z} \in \mathbb{R}^{|\mathcal{V}| \times |\mathcal{R}|C}$. At this time, each node is represented by a $|\mathcal{R}|C$-dimensional vector, *i.e.*, the aggregated messages of all relations. Next, we concatenate the convolutional kernel matrices of all relations to produce $W^{\text{conv}} \in \mathbb{R}^{|\mathcal{R}|C \times C}$, and this matrix is applied upon $\tilde{Z}$ to combine the messages in the same relational slot and aggregate messages across different relations:

$$Z^{\text{aggr}} = \tilde{Z}W^{\text{conv}}, \tag{8}$$

where $Z^{\text{aggr}} \in \mathbb{R}^{|\mathcal{V}| \times C}$ denotes the aggregated neighborhood information for each node. This step has the FLOPs as below:

$$\text{FLOPs}(\text{RGConv}-②) = 2|\mathcal{R}||\mathcal{V}|C^2. \tag{9}$$

③ In the final step, a self-update with matrix $W^{\text{self}} \in \mathbb{R}^{C \times C}$ is first performed on the input feature of each node, and the self-updated node feature is further added with the aggregated neighborhood information:

$$Z' = ZW^{\text{self}} + Z^{\text{aggr}}, \tag{10}$$

where $Z' \in \mathbb{R}^{|\mathcal{V}| \times C}$ denotes output node features. This step has the FLOPs as below:

$$\text{FLOPs}(\text{RGConv}-③) = 2|\mathcal{V}|C^2 + |\mathcal{V}|C. \tag{11}$$

Therefore, by summing up the computational cost of three steps, the RGConv consumes the following FLOPs in total:

$$\text{FLOPs}(\text{RGConv}) = |\mathcal{R}| \cdot (2\bar{d}|\mathcal{V}|C + 2|\mathcal{V}|C^2) + 2|\mathcal{V}|C^2 + |\mathcal{V}|C.$$

$\square$

**Proposition 2.** *To process the assumed multi-relational graph, the Gated Relational Message Passing (GRMP) consumes the FLOPs as below under the efficient implementation with sparse matrix multiplication:*

$$\text{FLOPs}(\text{GRMP}) = |\mathcal{R}| \cdot (2\bar{d} + 7)|\mathcal{V}|C + 6|\mathcal{V}|C^2.$$

*Proof.* Following the steps of GRMP stated in Eq. (3), we compute the FLOPs of each step:

① In the first step, we conduct a pre-layer node-wise channel aggregation with the weight matrix $W^{\text{in}} \in \mathbb{R}^{C \times C}$:

$$Z^{\text{in}} = ZW^{\text{in}}, \tag{12}$$

where $Z \in \mathbb{R}^{|\mathcal{V}| \times C}$ denotes the input node features, and $Z^{\text{in}} \in \mathbb{R}^{|\mathcal{V}| \times C}$ denotes the channel-aggregated node features. This step has the FLOPs consumption as below:

$$\text{FLOPs}(\text{GRMP}-①) = 2|\mathcal{V}|C^2. \tag{13}$$

② In the second step, we first gather the messages within the same relation for each node, which is realized by the sparse matrix multiplication between $Z^{\text{in}}$ and the adjacency matrix $A \in \mathbb{R}^{|\mathcal{V}| \times |\mathcal{R}||\mathcal{V}|}$ ($A$ is identically defined as in the step ① of Proposition 1):

$$\tilde{Z}^{\text{in}} = A^\top Z^{\text{in}}, \tag{14}$$

where $\tilde{Z}^{\text{in}} \in \mathbb{R}^{|\mathcal{R}||\mathcal{V}| \times C}$ represents the relational slots of all nodes after message passing. The relational slots of each node are then integrated to get the reshaped $\tilde{Z}^{\text{in}} \in \mathbb{R}^{|\mathcal{V}| \times |\mathcal{R}|C}$. By concatenating the convolutional kernel vectors of all relations, we have $w_{\text{conv}} \in \mathbb{R}^{|\mathcal{R}|C \times 1}$, and this vector is broadcast to all nodes to perform channel-wise message aggregation via Hadamard product:

$$\tilde{Z}^{\text{aggr}} = (\mathbf{1}_{\text{conv}} w_{\text{conv}}^\top) \odot \tilde{Z}^{\text{in}}, \tag{15}$$

where $\mathbf{1}_{\text{conv}} \in \mathbb{R}^{|\mathcal{V}| \times 1}$ is the all-one vector for broadcasting, and $\tilde{Z}^{\text{aggr}} \in \mathbb{R}^{|\mathcal{V}| \times |\mathcal{R}|C}$ denotes the relational slots of all nodes after intra-relation message aggregation.

To conduct the operations in Eqs. (14) and (15), this step consumes the following FLOPs:

$$\text{FLOPs}(\text{GRMP}-②) = 2|\mathcal{E}|C + 2|\mathcal{R}||\mathcal{V}|C = 2\bar{d}|\mathcal{R}||\mathcal{V}|C + 2|\mathcal{R}||\mathcal{V}|C. \tag{16}$$

③ In the third step, we first compute the attentive weights assigned to all relations on each node:

$$M^\alpha = ZW^\alpha, \tag{17}$$

where $W^\alpha \in \mathbb{R}^{C \times |\mathcal{R}|}$ is the weight matrix for node-adaptive relation weighting, and $M^\alpha \in \mathbb{R}^{|\mathcal{V}| \times |\mathcal{R}|}$ denotes the relation weights on all nodes. After that, a weighted summation is performed to aggregate the messages of different relations in $\tilde{Z}^{\text{aggr}}$ (in this operation, we use the reshaped $\tilde{Z}^{\text{aggr}} \in \mathbb{R}^{|\mathcal{V}| \times |\mathcal{R}| \times C}$ and the reshaped $M^\alpha \in \mathbb{R}^{|\mathcal{V}| \times |\mathcal{R}| \times 1}$):

$$\widehat{Z}^{\text{aggr}} = \sum_{i=1}^{|\mathcal{R}|} (M^\alpha_{:,i,:} \mathbf{1}_\alpha^\top) \odot \tilde{Z}^{\text{aggr}}_{:,i,:}, \tag{18}$$

where $\mathbf{1}_\alpha \in \mathbb{R}^{C \times 1}$ is the all-one vector for broadcasting relation weights to all feature channels, and $\widehat{Z}^{\text{aggr}} \in \mathbb{R}^{|\mathcal{V}| \times C}$ denotes the per-node neighborhood representations after inter-relation message aggregation.

To perform Eqs. (17) and (18), this step has the following FLOPs consumption:

$$\text{FLOPs}(\text{GRMP}-③) = 2|\mathcal{R}||\mathcal{V}|C + |\mathcal{R}| \cdot 2|\mathcal{V}|C + (|\mathcal{R}| - 1)|\mathcal{V}|C = 5|\mathcal{R}||\mathcal{V}|C - |\mathcal{V}|C. \tag{19}$$

④ The fourth step conducts a post-layer node-wise channel aggregation with the weight matrix $W^{\text{out}} \in \mathbb{R}^{C \times C}$:

$$Z^{\text{aggr}} = \widehat{Z}^{\text{aggr}} W^{\text{out}}, \tag{20}$$

where $Z^{\text{aggr}} \in \mathbb{R}^{|\mathcal{V}| \times C}$ denotes the channel-aggregated neighborhood representations. This step consumes the FLOPs as below:

$$\text{FLOPs}(\text{GRMP}-④) = 2|\mathcal{V}|C^2. \tag{21}$$

⑤ In the final step, the input feature of each node first performs self-update with the weight matrix $W^{\text{self}} \in \mathbb{R}^{C \times C}$, and the self-updated node feature is further updated by its neighborhood representation via a gating mechanism:

$$Z' = ZW^{\text{self}} \odot Z^{\text{aggr}}, \tag{22}$$

where $Z' \in \mathbb{R}^{|\mathcal{V}| \times C}$ denotes output node features. This step has the FLOPs as below:

$$\text{FLOPs}(\text{GRMP}-⑤) = 2|\mathcal{V}|C^2 + |\mathcal{V}|C. \tag{23}$$

Therefore, by summing up the computational cost of five steps, the GRMP has the following FLOPs consumption in total:

$$\text{FLOPs}(\text{GRMP}) = |\mathcal{R}| \cdot (2\bar{d} + 7)|\mathcal{V}|C + 6|\mathcal{V}|C^2. \tag{24}$$

□

# B  ANALYSIS OF MODEL EXPRESSIVITY

In this section, we study the expressivity of the proposed GRMP layer (Sec. 3.3). Specifically, we introduce the variant of the Weisfeiler-Leman (WL) algorithm (Morris et al., 2019) on multi-relational graphs and show that there exists parameterization of GRMP that is as expressive as the multi-relational WL algorithm.

Following the philosophy of the 1-dimensional Weisfeiler-Leman (1-WL) algorithm (Morris et al., 2019), we define the multi-relational 1-WL (1-RWL) algorithm. This algorithm studies a labeled multi-relational graph $\mathcal{G} = (\mathcal{V}, \mathcal{E}_1, \ldots, \mathcal{E}_{|\mathcal{R}|}, l)$, where $\mathcal{V}$ is the node set, $\mathcal{E}_i$ denotes the edge set associated with the $i$-th relation, and $l$ is the label function that assigns initial node features. The 1-RWL computes a node coloring $C^{(t)} : \mathcal{V} \to \mathbb{N}$ for each iteration $t \geqslant 0$, and the initial coloring $C^{(0)}$ is consistent with the label function $l$ (*i.e.*, one unique color for the nodes with a specific label). For iteration $t > 0$, 1-RWL updates the color of each node $v \in \mathcal{V}$ based on the colors of itself and its neighbors with different relations in the last iteration:

$$C^{(t)}(v) := \text{HASH}\Big(\big(C^{(t-1)}(v), \{\!\{(C^{(t-1)}(u), i) | i \in [|\mathcal{R}|], u \in \mathcal{N}_i(v)\}\!\}\big)\Big), \quad \forall v \in \mathcal{V}, \tag{25}$$

where $[|\mathcal{R}|] = \{1, \ldots, |\mathcal{R}|\}$ denotes the indices of all relations, $\mathcal{N}_i(v)$ is the neighborhood set of node $v$ with the $i$-th relation, $\{\!\{\ldots\}\!\}$ denotes a multiset. To test the isomorphism of two multi-relational graphs $\mathcal{G}$ and $\mathcal{G}'$, the 1-RWL algorithm is run in parallel on both graphs. If the number of nodes assigned with a specific color is different across two graphs at an iteration, it is concluded that $\mathcal{G}$ and $\mathcal{G}'$ are non-isomorphic. The algorithm terminates when the color assignments do not change across two iterations, which is reached after at most $\max\{|\mathcal{V}|, |\mathcal{V}'|\}$ iterations ($\mathcal{V}$ and $\mathcal{V}'$ are the node sets of two graphs). Just as the 1-WL test (Cai et al., 1992), the same color assignments along the whole process of 1-RWL cannot guarantee the isomorphism of two graphs, while it is still a powerful heuristic for (1) distinguishing the nodes with different structural roles in a multi-relational graph and (2) distinguishing non-isomorphic multi-relational graphs (Babai & Kucera, 1979).

We next compare the expressivity between the GRMP layer and the 1-RWL algorithm. For multi-relational graph $\mathcal{G}$, we denote $\mathbf{F} \in \mathbb{R}^{|\mathcal{V}| \times d}$ as the node feature matrix and $\mathbf{A}_i \in \mathbb{R}^{|\mathcal{V}| \times |\mathcal{V}|}$ as the adjacency matrix for the $i$-th relation ($i \in [|\mathcal{R}|]$). The node feature update rule of GRMP can be written as below:

$$\mathbf{F}' = (\mathbf{F}\mathbf{W}_{\text{self}} + \mathbf{b}_{\text{self}}) \odot \left(\Big(\sum\nolimits_{i \in [|\mathcal{R}|]} \alpha_i \mathbf{A}_i (\mathbf{F}\mathbf{W}_{\text{in}} + \mathbf{b}_{\text{in}})\Big)\mathbf{W}_{\text{out}} + \mathbf{b}_{\text{out}}\right), \tag{26}$$

where $\mathbf{W}_{\text{self}}$, $\mathbf{W}_{\text{in}}$ and $\mathbf{W}_{\text{out}}$ are the parameter matrices (*i.e.*, weights), $\mathbf{b}_{\text{self}}$, $\mathbf{b}_{\text{in}}$ and $\mathbf{b}_{\text{out}}$ are the parameter vectors (*i.e.*, biases), and $\alpha_i$ ($i \in [|\mathcal{R}|]$) are the per-relation scaling factors. Note that, in this analysis, we simplify GRMP's intra- and inter-relation message aggregation (step ② and step ③ in Eq. (3)) as per-relation scaling, where the expressivity of this simplified GRMP is upper bounded by the original one. Following Morris et al. (2019), we consider the model with a stack of GRMP layers and denote the sequence of GRMP parameters $\mathbf{W}_{\text{GRMP}}^{(t)}$ up to the $t$-th layer as below:

$$\mathbf{W}_{\text{GRMP}}^{(t)} = \big(\mathbf{W}_{\text{self}}^{(t')}, \mathbf{W}_{\text{in}}^{(t')}, \mathbf{W}_{\text{out}}^{(t')}, \mathbf{b}_{\text{self}}^{(t')}, \mathbf{b}_{\text{in}}^{(t')}, \mathbf{b}_{\text{out}}^{(t')}, \alpha_i\big)_{t' \leqslant t, i \in [|\mathcal{R}|]}. \tag{27}$$

On such basis, we next illustrate there exists parameters $\mathbf{W}_{\mathrm{GRMP}}^{(t)}$ such that the corresponding model is as expressive as the coloring $C^{(t)}$ in terms of distinguishing nodes in multi-relational graphs.

**Theorem 1.** *Let $\mathcal{G} = (\mathcal{V}, \mathcal{E}_1, \ldots, \mathcal{E}_{|\mathcal{R}|}, l)$ be a labeled multi-relational graph. For all $t \geqslant 0$, there exists initial node features and a sequence $\mathbf{W}_{\mathrm{GRMP}}^{(t)}$ of GRMP parameters such that the following holds:*

$$C^{(t)}(v) = C^{(t)}(w) \iff \mathbf{F}_v^{(t)} = \mathbf{F}_w^{(t)}, \quad \forall v, w \in \mathcal{V}.$$

*In other words, the node feature matrix $\mathbf{F}^{(t)}$ is equivalent to the coloring function $C^{(t)}$ of 1-RWL at all iterations.*

*Proof.* (1) **Prerequisites.** Following Morris et al. (2019), a matrix is denoted as *row-independent modulo equality* if the set of all different rows in the matrix are linearly independent. For two coloring functions $C_1$ and $C_2$ of $\mathcal{G}$, we denote their equivalence $C_1 \equiv C_2$ if they define the same partition over the node set $\mathcal{V}$. We next prove the result by induction.

(2) **Base case.** For $t = 0$, we define the initial node features $\mathbf{F}^{(0)}$ to be row-independent modulo equality and consistent with the label function $l$ (*e.g.*, the one-hot encoding of node labels). Since the initial coloring function satisfies $C^{(0)} = l$, we can conclude the equivalence of $C^{(0)}$ and $\mathbf{F}^{(0)}$, *i.e.*, $C^{(0)}(v) = C^{(0)}(w) \Leftrightarrow \mathbf{F}_v^{(0)} = \mathbf{F}_w^{(0)}, \forall v, w \in \mathcal{V}$.

(3) **Induction step.** For $t \geqslant 0$, we assume that $C^{(t)}$ and $\mathbf{F}^{(t)}$ are equivalent, and $\mathbf{F}^{(t)}$ is row-independent modulo equality. The coloring $C^{(t+1)}$ of 1-RWL at iteration $t+1$ is derived by applying a 1-RWL step to update the coloring $C^{(t)}$:

$$C^{(t+1)}(v) := \mathrm{HASH}\Big( \big( C^{(t)}(v), \{\!\{ (C^{(t)}(u), i) | i \in [|\mathcal{R}|], u \in \mathcal{N}_i(v) \}\!\} \big) \Big), \quad \forall v \in \mathcal{V}. \tag{28}$$

Let $q$ be the number of different colors defined by $C^{(t)}$ and let $Q_1, \ldots, Q_q$ be the $q$ different node subsets partitioned by $C^{(t)}$. The updated coloring $C^{(t+1)}$ can be equivalently represented by the matrix $\mathbf{D} \in \mathbb{R}^{|\mathcal{V}| \times q(|\mathcal{R}|+1)}$ with following entries:

$$\mathbf{D}_{vk} = \begin{cases} |\mathcal{N}_i(v) \cap Q_j| & \text{if } k = iq + j \text{ for } i \in [|\mathcal{R}|], j \in [q], \\ 1 & \text{if } k \in [q] \text{ and } v \in Q_k, \\ 0 & \text{otherwise,} \end{cases} \tag{29}$$

in which the corresponding row of node $v$ is the concatenation of an one-hot encoding of $v$'s color and a vector encoding for the multiset of the colors in $\mathcal{N}_i(v)$, for each $i \in [|\mathcal{R}|]$. Based on the row encoding of nodes in $\mathbf{D}$, we can partition the node set $\mathcal{V}$ into subsets (the nodes in the same subset share the same row encoding) and assign a unique color to each subset, which defines the coloring function $C_{\mathbf{D}}$. The equivalence $C_{\mathbf{D}} \equiv C^{(t+1)}$ holds.

Since $C^{(t)}$ and $\mathbf{F}^{(t)}$ are assumed to be equivalent, there should be $q$ distinct rows in $\mathbf{F}^{(t)}$, and each of them corresponds to one of $q$ different colors defined by $C^{(t)}$. Let $\tilde{\mathbf{F}}^{(t)} \in \mathbb{R}^{q \times d}$ be the matrix composed of these distinct rows with the order corresponding to $Q_1, \ldots, Q_q$. By assumption, the rows of $\tilde{\mathbf{F}}^{(t)}$ are linearly independent, and thus there is a matrix $\mathbf{M} \in \mathbb{R}^{d \times q}$ such that $\tilde{\mathbf{F}}^{(t)} \mathbf{M} \in \mathbb{R}^{q \times q}$ is an identity matrix. By extension, the matrix $\mathbf{F}^{(t)} \mathbf{M} \in \mathbb{R}^{|\mathcal{V}| \times q}$ has entries:

$$(\mathbf{F}^{(t)} \mathbf{M})_{vj} = \begin{cases} 1 & \text{if } v \in Q_j, \\ 0 & \text{otherwise.} \end{cases} \tag{30}$$

Note that, the matrix $\mathbf{D}$ defined in Eq. (29) can be viewed as a block matrix $\mathbf{D} = [\mathbf{B}_0 \, \mathbf{B}_1 \, \ldots \, \mathbf{B}_{|\mathcal{R}|}]$, where $\mathbf{B}_0 = \mathbf{F}^{(t)} \mathbf{M} \in \mathbb{N}^{|\mathcal{V}| \times q}$ and $\mathbf{B}_i = \mathbf{A}_i \mathbf{F}^{(t)} \mathbf{M} \in \mathbb{N}^{|\mathcal{V}| \times q}$ for each $i \in [|\mathcal{R}|]$. Since each element of $\mathbf{D}$ is upper bounded by $|\mathcal{V}| - 1$, we follow polynomial coding to assign $\mathbf{B}_i$ ($i \in [|\mathcal{R}|]$) with the polynomial term $|\mathcal{V}|^i$ and assign $\mathbf{B}_0$ with the term $|\mathcal{V}|^{|\mathcal{R}|+1}$, defining the following matrix $\mathbf{E} \in \mathbb{N}^{|\mathcal{V}| \times q}$ which is equivalent to $\mathbf{D}$ in terms of partitioning nodes based on the rows of matrix:

$$\mathbf{E} = \big( |\mathcal{V}|^{|\mathcal{R}|+1} \cdot \mathbf{F}^{(t)} \mathbf{M} + \mathbb{1} \big) \odot \Big( \sum\nolimits_{i \in [|\mathcal{R}|]} |\mathcal{V}|^i \cdot \mathbf{A}_i \mathbf{F}^{(t)} \mathbf{M} + \mathbb{1} \Big), \tag{31}$$

where $\mathbb{1} \in \mathbb{N}^{|\mathcal{V}| \times q}$ is an all-one matrix with fitted shape. The matrix $\mathbf{E}$ defines a coloring function $C_{\mathbf{E}}$ in the same way as the matrix $\mathbf{D}$, and the equivalence $C_{\mathbf{E}} \equiv C_{\mathbf{D}}$ holds.

By aligning the update rule of GRMP (Eq. (26)) with the definition of matrix $\mathbf{E}$ (Eq. (31)), we adopt the parameterization $\mathbf{W}_{\text{self}} = |\mathcal{V}|^{|\mathcal{R}|+1}\mathbf{M}$, $\mathbf{W}_{\text{in}} = \mathbf{M}$, $\mathbf{W}_{\text{out}} = \mathbf{I}_q$ (the $q \times q$ identity matrix), $\mathbf{b}_{\text{self}} = \mathbb{1}$, $\mathbf{b}_{\text{in}} = \mathbf{0}_{|\mathcal{V}|,q}$ (the $|\mathcal{V}| \times q$ zero matrix), $\mathbf{b}_{\text{out}} = \mathbb{1}$, and $\alpha_i = |\mathcal{V}|^i$. In this way, the node feature matrix $\mathbf{F}^{(t+1)}$ at iteration $t+1$ is updated from $\mathbf{F}^{(t)}$ as below:

$$\mathbf{F}^{(t+1)} = \left(|\mathcal{V}|^{|\mathcal{R}|+1} \cdot \mathbf{F}^{(t)}\mathbf{M} + \mathbb{1}\right) \odot \left(\left(\sum\nolimits_{i \in [|\mathcal{R}|]} |\mathcal{V}|^i \cdot \mathbf{A}_i(\mathbf{F}^{(t)}\mathbf{M} + \mathbf{0}_{|\mathcal{V}|,q})\right)\mathbf{I}_q + \mathbb{1}\right) = \mathbf{E}, \quad (32)$$

Therefore, the coloring function $C_{\mathbf{F}^{(t+1)}}$ defined by the updated node feature matrix $\mathbf{F}^{(t+1)}$ satisfies:

$$C_{\mathbf{F}^{(t+1)}} \equiv C_{\mathbf{E}} \equiv C_{\mathbf{D}} \equiv C^{(t+1)}. \quad (33)$$

In particular, we have: $C^{(t+1)}(v) = C^{(t+1)}(w) \Leftrightarrow \mathbf{F}_v^{(t+1)} = \mathbf{F}_w^{(t+1)}, \forall v, w \in \mathcal{V}$.

(4) **Conclusion.** Since both the base case and the induction step have been proved to be true, we can conclude that:

$$C^{(t)}(v) = C^{(t)}(w) \iff \mathbf{F}_v^{(t)} = \mathbf{F}_w^{(t)}, \quad \forall t \geqslant 0, \forall v, w \in \mathcal{V}. \quad (34)$$

$\square$

This result proves the equivalent expressivity of the 1-RWL algorithm and the model constructed by simplified GRMP layers (Eq. (26)). Therefore, when constructing the model with standard GRMP layers (Eq. (3)), the model is at least as expressive as the 1-RWL algorithm.

# C GRAPHICAL ILLUSTRATION OF GRMP

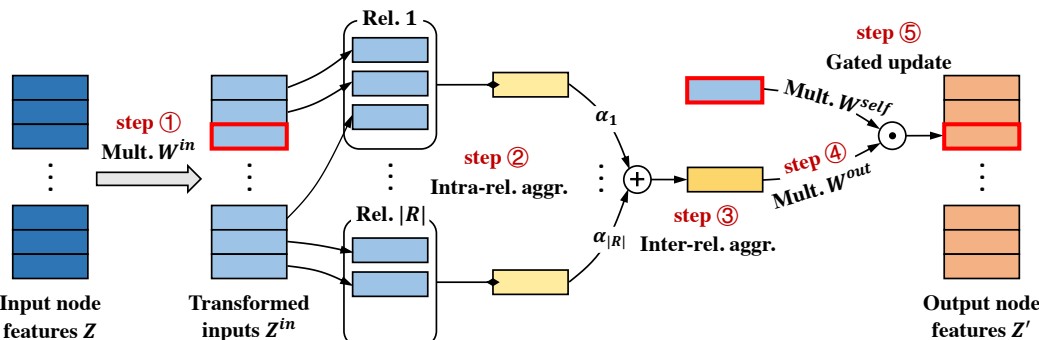

Figure 5: Graphical illustration for node representation update in the GRMP layer. We specifically show the neighborhood aggregation and representation update procedure of the node denoted in red. *Abbr.*, Multi.: multiply with; Rel.: relation; aggr.: aggregation.

In Fig. 5, we graphically illustrate the mechanism of node representation update in the GRMP layer. In specific, GRMP updates the node representation matrix from $Z$ to $Z'$ with the following steps:

① A linear layer transforms the input node representations $Z \in \mathbb{R}^{|\mathcal{V}| \times C}$ to $Z^{\text{in}} \in \mathbb{R}^{|\mathcal{V}| \times C}$, which aggregates the feature channels of each node at the beginning of the layer.

② For each node, its neighbors are assigned to different groups according to their relations with the node, and the neighbors in each group are aggregated in a channel-wise way.

③ The aggregated messages of different relational groups are then scaled by per-relation scalar weights $\{\alpha_r\}_{r=1}^{|\mathcal{R}|}$ and aggregated to the neighborhood representations $\widehat{Z}^{\text{aggr}} \in \mathbb{R}^{|\mathcal{V}| \times C}$.

④ $\widehat{Z}^{\text{aggr}}$ is then transformed by a linear layer to aggregate the feature channels of each node's neighbors, deriving the transformed neighborhood representations $Z^{\text{aggr}} \in \mathbb{R}^{|\mathcal{V}| \times C}$.

⑤ Finally, $Z^{\text{aggr}}$ serves as the gate to update all node representations, deriving the output node representations $Z' \in \mathbb{R}^{|\mathcal{V}| \times C}$.

Table 7: Detailed architectures of EurNet-T/S/B for ImageNet-1K classification (#parameters and FLOPs are computed under the resolution $224 \times 224$). $H \times W$: input image resolution; $C$: number of feature channels; $\gamma$: FFN's hidden dimension ratio; $K$: number of K-nearest neighbors for medium-range edges; $\mathcal{Y}$: label set for classification. "T" denotes the tiny model; "S" denotes the small model; "B" denotes the base model.

| Module | #Patches | EurNet-T | EurNet-S | EurNet-B |
|---|---|---|---|---|
| **Stem** | $\frac{H}{4} \times \frac{W}{4}$ | $4 \times 4$ conv, stride$=4$ | $4 \times 4$ conv, stride$=4$ | $4 \times 4$ conv, stride$=4$ |
| **Graph Construction** | $\frac{H}{4} \times \frac{W}{4}$ | $\begin{bmatrix} \text{short-range edges,} \\ \text{long-range edges} \end{bmatrix}$ | $\begin{bmatrix} \text{short-range edges,} \\ \text{long-range edges} \end{bmatrix}$ | $\begin{bmatrix} \text{short-range edges,} \\ \text{long-range edges} \end{bmatrix}$ |
| **Stage 1** | $\frac{H}{4} \times \frac{W}{4}$ | $\begin{bmatrix} \text{GRMP} (C=96), \\ \text{FFN} (C=96, \gamma=4) \end{bmatrix} \times 2$ | $\begin{bmatrix} \text{GRMP} (C=96), \\ \text{FFN} (C=96, \gamma=4) \end{bmatrix} \times 2$ | $\begin{bmatrix} \text{GRMP} (C=128), \\ \text{FFN} (C=128, \gamma=4) \end{bmatrix} \times 2$ |
| **Downsample** | $\frac{H}{8} \times \frac{W}{8}$ | PatchMerging | PatchMerging | PatchMerging |
| **Graph Construction** | $\frac{H}{8} \times \frac{W}{8}$ | $\begin{bmatrix} \text{short-range edges,} \\ \text{medium-range edges} (K=12), \\ \text{long-range edges} \end{bmatrix}$ | $\begin{bmatrix} \text{short-range edges,} \\ \text{medium-range edges} (K=12), \\ \text{long-range edges} \end{bmatrix}$ | $\begin{bmatrix} \text{short-range edges,} \\ \text{medium-range edges} (K=12), \\ \text{long-range edges} \end{bmatrix}$ |
| **Stage 2** | $\frac{H}{8} \times \frac{W}{8}$ | $\begin{bmatrix} \text{GRMP} (C=192), \\ \text{FFN} (C=192, \gamma=4) \end{bmatrix} \times 2$ | $\begin{bmatrix} \text{GRMP} (C=192), \\ \text{FFN} (C=192, \gamma=4) \end{bmatrix} \times 2$ | $\begin{bmatrix} \text{GRMP} (C=256), \\ \text{FFN} (C=256, \gamma=4) \end{bmatrix} \times 2$ |
| **Downsample** | $\frac{H}{16} \times \frac{W}{16}$ | PatchMerging | PatchMerging | PatchMerging |
| **Graph Construction** | $\frac{H}{16} \times \frac{W}{16}$ | $\begin{bmatrix} \text{short-range edges,} \\ \text{medium-range edges} (K=12), \\ \text{long-range edges} \end{bmatrix}$ | $\begin{bmatrix} \text{short-range edges,} \\ \text{medium-range edges} (K=12), \\ \text{long-range edges} \end{bmatrix}$ | $\begin{bmatrix} \text{short-range edges,} \\ \text{medium-range edges} (K=12), \\ \text{long-range edges} \end{bmatrix}$ |
| **Stage 3** | $\frac{H}{16} \times \frac{W}{16}$ | $\begin{bmatrix} \text{GRMP} (C=384), \\ \text{FFN} (C=384, \gamma=4) \end{bmatrix} \times 6$ | $\begin{bmatrix} \text{GRMP} (C=384), \\ \text{FFN} (C=384, \gamma=4) \end{bmatrix} \times 18$ | $\begin{bmatrix} \text{GRMP} (C=512), \\ \text{FFN} (C=512, \gamma=4) \end{bmatrix} \times 18$ |
| **Downsample** | $\frac{H}{32} \times \frac{W}{32}$ | PatchMerging | PatchMerging | PatchMerging |
| **Graph Construction** | $\frac{H}{32} \times \frac{W}{32}$ | $\begin{bmatrix} \text{short-range edges,} \\ \text{medium-range edges} (K=12), \\ \text{long-range edges} \end{bmatrix}$ | $\begin{bmatrix} \text{short-range edges,} \\ \text{medium-range edges} (K=12), \\ \text{long-range edges} \end{bmatrix}$ | $\begin{bmatrix} \text{short-range edges,} \\ \text{medium-range edges} (K=12), \\ \text{long-range edges} \end{bmatrix}$ |
| **Stage 4** | $\frac{H}{32} \times \frac{W}{32}$ | $\begin{bmatrix} \text{GRMP} (C=768), \\ \text{FFN} (C=768, \gamma=4) \end{bmatrix} \times 2$ | $\begin{bmatrix} \text{GRMP} (C=768), \\ \text{FFN} (C=768, \gamma=4) \end{bmatrix} \times 2$ | $\begin{bmatrix} \text{GRMP} (C=1024), \\ \text{FFN} (C=1024, \gamma=4) \end{bmatrix} \times 2$ |
| **Head** | $1 \times 1$ | Pooling & Linear ($\vert\mathcal{Y}\vert=1000$) | Pooling & Linear ($\vert\mathcal{Y}\vert=1000$) | Pooling & Linear ($\vert\mathcal{Y}\vert=1000$) |
| **#Parameters (M)** | | 28.8 | 50.2 | 88.7 |
| **FLOPs (G)** | | 4.6 | 8.8 | 15.6 |

# D   DETAILED MODEL ARCHITECTURE FOR IMAGE MODELING

For image modeling, we basically follow the hierarchical architecture proposed by Swin Transformer (Liu et al., 2021b), as summarized in Tab. 7. The architecture begins with a patch embedding module implemented by non-overlapping 2D convolution. After that, the model is split into 4 modeling stages: (1) the number of patches (*i.e.*, nodes in our graph) is reduced to a quarter across consecutive stages by the "PatchMerging" operation (Liu et al., 2021b); (2) increasing feature channels $[C, 2C, 4C, 8C]$ are used for all stages. We place a graph construction layer before each modeling stage to update the multi-relational graph structure. For the first stage, we only use short- and long-range edges to reduce the computational cost (computing medium-range edges by representation similarity comparison is expensive in the first stage with many patches), and the relational edges of all three ranges are adopted in the last three stages. Each stage is composed of multiple modeling blocks, where each block contains a GRMP layer (Sec. 3.3) for relational message passing and a feed-forward network (FFN) (Vaswani et al., 2017) for feature transformation. In the end, a global average pooling layer produces the whole-image representation, and a linear head outputs the final prediction. We adjust the number of feature channels and the number of blocks in each stage to derive EurNet-T, EurNet-S and EurNet-B with standard number of parameters and FLOPs. We implement the models based on the PyTorch (Paszke et al., 2017) deep learning library.

# E   INTRODUCTION TO PROTEIN STRUCTURE

Proteins are macromolecules that perform critical biological functions in living organisms. A protein owns multiple levels of structures, as described below:

- **Primary structure** (Fig. 6(a)). At the chemical level, a protein is composed of one or multiple chains of amino acid residues, forming the protein sequence which is the primary protein struc-

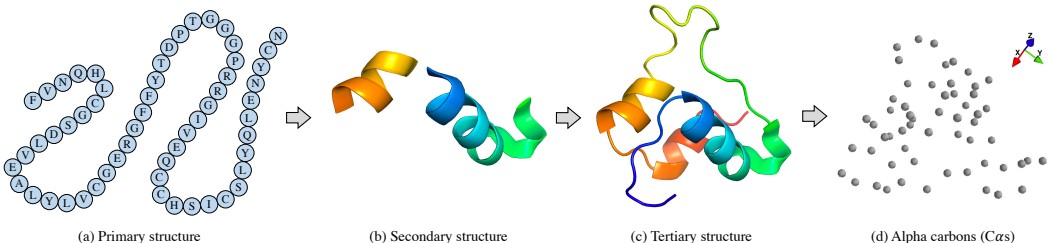

(a) Primary structure  (b) Secondary structure  (c) Tertiary structure  (d) Alpha carbons (Cαs)

Figure 6: The primary structure, secondary structure, tertiary structure and all alpha carbons of the single-chain insulin protein (ID in PDB (Berman et al., 2000): 2LWZ).

ture. In the protein sequence $s = (s_1, s_2, \cdots, s_L)$, each element $s_l$ denotes a type of amino acid (there are 20 common amino acids and two rare ones, *i.e.*, Selenocysteine and Pyrrolysine). The primary structure tells the sequential order of amino acids in a protein, but otherwise it does not reveal any information about the 3D folded structure of the protein. This fact limits its usefulness in the analysis/prediction of protein functions, due to the principle that "protein folded structures largely determine their functions" (Harms & Thornton, 2010).

- **Secondary structure** (Fig. 6(b)). The secondary structures of proteins are some repeatedly-occurred local structures like the $\alpha$-helices shown in Fig. 6(b). These structures are stabilized by hydrogen bonds, and, together with the tight turns and flexible loops in between, they constitute the complete protein folded structure.

- **Tertiary structure** (Fig. 6(c)). The spatial arrangement of different secondary structure components leads to the formation of the tertiary structure (*i.e.*, the folded structure of a protein). The tertiary structure is jointly held by short-range interactions like hydrogen bonding and long-range interactions like hydrophobic interactions. Thanks to the recent advances of highly accurate protein folded structure predictors based on deep learning (Jumper et al., 2021; Baek et al., 2021), we can now efficiently acquire numerous previously unknown protein tertiary structures with reasonable confidence. These advances are expected to promote the understanding of protein functions based on tertiary structures.

In this work, we focus on protein function prediction tasks based on tertiary structures. Specifically, we adopt an informative and light-weight representation format, *i.e.*, **all alpha carbons (Cαs) in the tertiary structure** (Fig. 6(d)), which is widely used in the literature (Gligorijević et al., 2021; Baldassarre et al., 2021; Zhang et al., 2022). A C$\alpha$ can be seen as the center of its corresponding amino acid, and thus the overall tertiary structure of a protein can be well captured by the collection of all C$\alpha$s. At this time, the C$\alpha$s are actually a set of separate points in the 3D space, since there is no chemical bond among them. To better describe the interactions within a protein, we seek to construct edges among C$\alpha$s and lead to a more informative representation format, *i.e.*, the **C$\alpha$ graph**.

## F  MORE EXPERIMENTAL SETUPS

### F.1  MORE EXPERIMENTAL SETUPS ON IMAGENET-1K CLASSIFICATION

In the following, we state the detailed model and training configurations of (1) training on ImageNet-1K from scratch and (2) pretraining on ImageNet-22K followed by ImageNet-1K fine-tuning. For training configurations, we mainly follow the standards set up by Swin Transformer (Liu et al., 2021b) for fair comparison.

### F.1.1  FROM-SCRATCH TRAINING ON IMAGENET-1K

**Model configurations.** The whole model architectures of EurNet-T, EurNet-S and EurNet-B are presented in Tab. 7. For medium-range edges, 12 nearest semantic neighbors of each patch are linked to it to capture medium-range interactions. For long-range edges, we compute the representations of per-patch global-context virtual nodes by a stack of depth-wise 2D convolutions with the accumulative receptive field as 7, and these virtual nodes are linked to their corresponding patches.

**Training configuration.** An AdamW (Loshchilov & Hutter, 2017) optimizer (betas: [0.9, 0.999], weight decay: 0.05) is employed to train each EurNet model for 300 epochs. We set the batch size as 2048, the base learning rate as 0.002 and the gradient clipping norm as 5.0. A cosine learning rate scheduler is adopted to adjust the learning rate from $2.0 \times 10^{-6}$ to 0.002 in the first 20 warm-up epochs, and the learning rate is decayed to $2.0 \times 10^{-5}$ in the rest epochs with a cosine rate. The stochastic depth drop rates are set to 0.15, 0.3 and 0.5 respectively for EurNet-T, EurNet-S and EurNet-B. We follow the augmentation functions and mixup strategies used in Swin Transformer. All experiments are conducted on 16 Tesla-V100-32GB GPUs.

### F.1.2 PRE-TRAINING ON IMAGENET-22K AND FINE-TUNING ON IMAGENET-1K

**Model configurations.** The EurNet-B with the standard model architecture as in Tab. 7 is used, except that the last linear classification head outputs 21,841-dimensional logits to perform ImageNet-22K classification.

**Training configuration.** For ImageNet-22K pre-training, we train EurNet-B with an AdamW optimizer (betas: [0.9, 0.999], weight decay: 0.05) for 90 epochs with the batch size 4096 and the image resolution $224 \times 224$. A cosine learning rate scheduler is employed to linearly increase the learning rate from 0 to $4.0 \times 10^{-3}$ in the first 5 warm-up epochs, and it decays the learning rate to $1.0 \times 10^{-6}$ in the rest epochs with a cosine rate. The stochastic depth drop rate is set as 0.1. All augmentation functions and mixup strategies follow Swin Transformer. The pre-training is performed on 64 Tesla-V100-32GB GPUs.

For ImageNet-1K fine-tuning, the pre-trained model is fine-tuned for 30 epochs by an AdamW optimizer (betas: [0.9, 0.999], weight decay: $1.0 \times 10^{-8}$). The cosine learning rate scheduler adjusts the learning rate from $8.0 \times 10^{-8}$ to $8.0 \times 10^{-5}$ in the first 5 warm-up epochs, and the learning rate is decayed to $8.0 \times 10^{-7}$ in the rest epochs with a cosine rate. The stochastic depth drop rate is set as 0.2. Both Mixup (Zhang et al., 2017) and CutMix (Yun et al., 2019) are muted during fine-tuning, following FocalNet Yang et al. (2022). The fine-tuning is performed on 16 Tesla-V100-32GB GPUs.

### F.1.3 THROUGHPUT COMPUTATION

We follow Swin Transformer to measure the inference throughput on a Tesla-V100-32GB GPU with batch size 128. We adopt graph checkpoints to enhance the speed of inferring an image that has been seen. During inference, we add short-range edges to the list of medium-range edges and merge their corresponding relations to further promote the efficiency.

### F.2 MORE EXPERIMENTAL SETUPS ON COCO OBJECT DETECTION

**Model configurations.** We use the EurNet-T, EurNet-S and EurNet-B pre-trained on ImageNet-1K as the backbone of Mask R-CNN (He et al., 2017). In specific, we take the patch representations output from all four modeling stages as the inputs of the Feature Pyramid Network (FPN) Lin et al. (2017). For medium-range edge construction on the high-resolution images of COCO, we select the semantic neighbors of each patch from a $112 \times 112$ dilated window (dilation ratio: 2) to reduce the computational cost. For long-range edge construction, the representations of per-patch global-context virtual nodes are computed by a stack of depth-wise 2D convolutions with the accumulative receptive field as 31, and these virtual nodes are linked to their corresponding patches.

**Training configurations.** We follow Swin Transformer (Liu et al., 2021b) to adopt a multi-scale training strategy where the shorter side of an image is resized to $[480, 800]$, and the longer side is with length 1,333. An AdamW (Loshchilov & Hutter, 2017) optimizer (betas: [0.9, 0.999], weight decay: 0.05) with initial learning rate $1.0 \times 10^{-4}$ is employed for model training. In the 1× schedule with 12 total epochs, the learning rate is decayed at the 9th and 11th epoch with the decay rate 0.1. In the 3× schedule with 36 total epochs, the learning rate is decayed at the 27th and 33rd epoch with the decay rate 0.1. The stochastic depth drop rate is set as 0.1, 0.2, 0.3 in 1× schedule and 0.25, 0.5, 0.5 in 3× schedule for EurNet-T/S/B, respectively. All models are trained with batch size 8 on 8 Tesla-V100-32GB GPUs (*i.e.*, one image per GPU). Our implementations are based on the mmdetection (Chen et al., 2019a) framework.

### F.3 MORE EXPERIMENTAL SETUPS ON ADE20K SEMANTIC SEGMENTATION

**Model configurations.** The EurNet-T, EurNet-S and EurNet-B pre-trained on ImageNet-1K serve as the backbone of UperNet (Xiao et al., 2018) to perform semantic segmentation. The patch representations output by all four modeling stages serve as the inputs of the Feature Pyramid Network (FPN) Lin et al. (2017). For medium-range edge construction, each patch is connected with its semantic neighbors from a $144 \times 144$ dilated window (dilation ratio: 2). For long-range edge construction, we use a stack of depth-wise 2D convolutions with accumulative receptive field 31 to compute the representations of per-patch global-context virtual nodes, and we connect these virtual nodes with their corresponding patches.

**Training configurations.** All input images are resized to the resolution $512 \times 512$. We adopt an AdamW (Loshchilov & Hutter, 2017) optimizer (betas: [0.9, 0.999], weight decay: 0.01) to train the model for 160K iterations with the base learning rate $6.0 \times 10^{-5}$. All models are trained with batch size 16 on 8 Tesla-V100-32GB GPUs (*i.e.*, two images per GPU). Our implementations are based on the mmsegmentation (Contributors, 2020) framework.

### F.4 MORE EXPERIMENTAL SETUPS ON PROTEIN FUNCTION PREDICTION

**Edge message passing.** Zhang et al. (2022) proposes to enhance the GearNet by edge-level message passing, which well captures the interactions between edges. To compare with the GearNet-Edge model enhanced in this way, we adapt the same edge message passing scheme to our EurNet.

Specifically, based on the constructed multi-relational graph $\mathcal{G} = (\mathcal{V}, \mathcal{E}, \mathcal{R})$, we further construct a *line graph* (Harary & Norman, 1960) $\mathcal{G}_{\text{line}} = (\mathcal{V}_{\text{line}}, \mathcal{E}_{\text{line}}, \mathcal{R}_{\text{line}})$. In this graph, each node $v \in \mathcal{V}_{\text{line}}$ corresponds to an edge in the original graph $\mathcal{G}$. There will an edge $(u, v, r)$ between nodes $u, v \in \mathcal{V}_{\text{line}}$ if the corresponding edges of $u$ and $v$ are adjacent in the original graph, and the edge type $r \in \{0, 1, \cdots, 7\}$ is determined by the angle $\angle_{(u,v)}$'s allocation in 8 equally-divided bins of $[0, \pi]$ ($\angle_{(u,v)}$ denotes the angle between the corresponding edges of $u$ and $v$ in the original graph). Based on this multi-relational line graph, we employ the GRMP layer (Sec. 3.3) to propagate information between the nodes in $\mathcal{G}_{\text{line}}$ and thus between the edges in the original graph $\mathcal{G}$. Readers are referred to Zhang et al. (2022) for more details. We name the EurNet equipped with such an edge message passing scheme as EurNet-Edge.

**Dataset details.** Two standard protein function prediction benchmarks are used in our experiments:

- **Enzyme Commission (EC) number prediction** Gligorijević et al. (2021) requires the model to predict the EC numbers of a protein based on its tertiary structure, where the EC numbers describe the protein's catalysis of biochemical reactions. This task involves the binary prediction of 538 different EC numbers, forming 538 binary classification problems. This dataset contains 15,550 training, 1,729 validation and 1,919 test proteins.

- **Gene Ontology (GO) term prediction** (Gligorijević et al., 2021) seeks to predict the GO terms owning by a protein based on its tertiary structure. This benchmark is further split into three branches based on three types of ontologies: biological process (BP), molecular function (MF) and cellular component (CC). Each branch is formed by multiple binary classification problems. The GO benchmark dataset contains 29,898 training, 3,322 validation and 3,415 test proteins.

**Model configurations.** The backbone architecture of EurNet is described in Sec. 4.2.2. Based on this backbone, we append a three-layer MLP with the architecture $\text{Linear}(C_{\text{out}}, C_{\text{out}}) \to \text{ReLU} \to \text{Linear}(C_{\text{out}}, C_{\text{out}}) \to \text{ReLU} \to \text{Linear}(C_{\text{out}}, N_{\text{task}})$ to predict the binary classification logits of all tasks simultaneously ($C_{\text{out}}$: the dimension of output protein representation; $N_{\text{task}}$: the number of binary classification tasks). We employ the binary cross entropy loss for model optimization.

**Training configurations.** An AdamW (Loshchilov & Hutter, 2017) optimizer (betas: [0.9, 0.999], weight decay: 0) is utilized to train the model for 200 epochs. We adopt a cosine learning rate scheduler to linearly increase the learning rate from $1.0 \times 10^{-7}$ to $1.0 \times 10^{-4}$, and the learning rate is decayed to $1.0 \times 10^{-6}$ in the rest epochs with a cosine rate. All models are trained with batch size 16 on 4 Tesla-V100-32GB GPUs (*i.e.*, four proteins per GPU).

Table 8: Performance comparison on knowledge graph completion benchmarks. "↓" denotes the metric is the lower the better; "↑" denotes the metric is the higher the better.

| Class | Model | FB15k-237 | | | | | WN18RR | | | | |
|---|---|---|---|---|---|---|---|---|---|---|---|
| | | MR$_\downarrow$ | MRR$_\uparrow$ | H@1$_\uparrow$ | H@3$_\uparrow$ | H@10$_\uparrow$ | MR$_\downarrow$ | MRR$_\uparrow$ | H@1$_\uparrow$ | H@3$_\uparrow$ | H@10$_\uparrow$ |
| **Embedding** | TransE | 357 | 0.294 | - | - | 0.465 | 3384 | 0.226 | - | - | 0.501 |
| | DistMult | 254 | 0.241 | 0.155 | 0.263 | 0.419 | 5110 | 0.43 | 0.39 | 0.44 | 0.49 |
| | ComplEx | 339 | 0.247 | 0.158 | 0.275 | 0.428 | 5261 | 0.44 | 0.41 | 0.46 | 0.51 |
| | RotatE | 177 | 0.338 | 0.241 | 0.375 | 0.553 | 3340 | 0.476 | 0.428 | 0.492 | 0.571 |
| **GNN** | RGCN | 221 | 0.273 | 0.182 | 0.303 | 0.456 | 2719 | 0.402 | 0.345 | 0.437 | 0.494 |
| | CompGCN | 197 | 0.355 | 0.264 | 0.390 | 0.535 | 3533 | 0.479 | 0.443 | 0.494 | 0.546 |
| | EurNet | **126** | **0.374** | **0.276** | **0.415** | **0.571** | **680** | **0.527** | **0.472** | **0.547** | **0.636** |

# G    EURNET FOR KNOWLEDGE GRAPH COMPLETION

**Datasets.** We conduct experiments on two standard knowledge graphs, FB15k-237 (Toutanova & Chen, 2015) and WN18RR (Dettmers et al., 2018). FB15k-237 contains 14,541 entities, 237 relation, 272,115 training triplets, 17,535 validation triplets and 20,466 test triplets. WN18RR has 40,943 entities, 11 relations, 86,835 training triplets, 3,034 validation triplets and 3,134 test triplets. We follow the TorchDrug library (Zhu et al., 2022) to process knowledge graphs. For each triplet $<h, r, t>$, its flipped counterpart $<t, r^{-1}, h>$ is included for data augmentation. All triplets from the validation and test sets are removed to form the fact graph for training.

**Model architecture.** In this experiment, we instantiate the EurNet with 6 GRMP layers, each with 32 feature channels. Upon the EurNet, we adopt a two-layer MLP activated by ReLU to score each candidate triplet.

**Training and evaluation.** For model training, we follow the default setting in the TorchDrug library (Zhu et al., 2022) to sample 32 negative triplets for each positive triplet and perform binary classification with the binary cross entropy loss. On both knowledge graphs, the EurNet is trained for 20 epochs by an Adam optimizer with learning rate $5.0 \times 10^{-3}$ and batch size 16. Model training is performed on 4 Tesla-V100-32GB GPUs. For evaluation, we follow previous works Vashishth et al. (2019); Zhu et al. (2021) to report mean rank (MR), mean reciprocal rank (MRR) and HITS at N (H@N) for knowledge graph completion.

**Baselines.** We compare the proposed EurNet with four classical knowledge graph embedding methods, *i.e.*, TransE (Bordes et al., 2013), DistMult (Yang et al., 2014), ComplEx (Trouillon et al., 2016) and RotatE (Sun et al., 2019), and two typical relational GNNs, *i.e.*, RGCN (Schlichtkrull et al., 2018) and CompGCN (Vashishth et al., 2019).

**Results.** We present the performance of EurNet and baselines in Tab. 8. It can be observed that EurNet clearly outperforms the embedding-based and GNN baselines on all metrics of two datasets. Although knowledge graphs contain no spatial information, they are representative multi-relational graphs and are good test fields for evaluating the capacity of relational GNNs. The superior performance of EurNet on these benchmarks demonstrates the effectiveness of the GRMP layer on modeling the complex relational patterns in knowledge graphs.

# H    MORE ABLATION STUDY

## H.1    EFFECT OF GRMP COMPONENTS

In Tab. 9, we analyze the key components of GRMP by substituting or removing the original component. This part of ablation studies are conducted on ImageNet-1K classification with EurNet-T.

Table 9: Ablation study of the key components of GRMP on ImageNet-1K with EurNet-T.

| Setting | #Params. (M) | FLOPs (G) | Throughput (imgs/s) | Top-1 Acc (%) |
|---|---|---|---|---|
| GRMP | 28.8 | 4.6 | 530.3 | 82.3 |
| GRMP (gating → addition) | 28.8 | 4.6 | 530.3 | 81.6$_{(\downarrow 0.7)}$ |
| GRMP ($\alpha_r(v) \rightarrow |\mathcal{R}|^{-1}$) | 28.8 | 4.6 | 567.4 | 81.9$_{(\downarrow 0.4)}$ |
| GRMP (*w/o* $W^{\text{in}}$) | 26.7 | 4.3 | 561.9 | 81.7$_{(\downarrow 0.6)}$ |
| GRMP (*w/o* $W^{\text{out}}$) | 26.7 | 4.3 | 562.5 | 81.5$_{(\downarrow 0.8)}$ |

**Effect of gating mechanism.** In the first row of the second block, we study the importance of the gating mechanism in GRMP by substituting the Hadamard product in the step ⑤ of Eq. (3) with the addition. After such a change, the top-1 accuracy decays by 0.7%. This performance

decay demonstrates that, by using the separable graph convolution scheme in GRMP, the gating operation is more suitable than addition for node representation update (in contrast to the additive node representation update of RGConv in Eq. (1)), which shares similar insights with the modulation mechanism in FocalNet (Yang et al., 2022).

**Effect of node-adaptive relation weighting.** In the second row of the second block, we replace GRMP's node-adaptive relation weighting operation with simply taking the mean over all relations. This change leads to a 0.4% drop of accuracy. This relation weighting operation helps the GRMP layer to adaptively aggregate the messages of different relations based on each node's status, which benefits the model performance.

**Effect of pre-layer and post-layer node-wise channel aggregation.** In the third and fourth rows of the second block, we respectively evaluate the model variants without $W^{\text{in}}$ and $W^{\text{out}}$. Under these two settings, the model accuracy decays by 0.6% and 0.8%, respectively. Therefore, it is important to perform both pre-layer and post-layer node-wise channel aggregation in the GRMP layer.

## H.2 EFFECT OF MULTI-RANGE EDGES FOR PROTEIN STRUCTURE MODELING

Tab. 10 shows the performance of EurNet on the EC function prediction benchmark by using different ranges of edges. When a single range of edges are employed, the model with short-range edges obtains the highest $F_{\text{max}}$ score 0.750. This result illustrates the importance of capturing short-range interactions for protein structure modeling, which coincides with the fact that many short-range interactions (*e.g.*, peptide and hydrogen bonds) contribute to the formation of protein structures. By adding long-range edges, the model performance is improved to 0.760, where the extra modeling of long-range interactions (*e.g.*, hydrophobic interactions) con-

Table 10: Ablation study of multi-range edges on EC with EurNet.

| short | medium | long | $F_{\text{max}}$ |
|:---:|:---:|:---:|:---:|
| ✓ | | | **0.750** |
| | ✓ | | 0.708 |
| | | ✓ | 0.647 |
| ✓ | ✓ | | 0.755 |
| ✓ | | ✓ | **0.760** |
| | ✓ | ✓ | 0.720 |
| ✓ | ✓ | ✓ | **0.768** |

tributes to this improvement. By using all three ranges of edges, the full model of EurNet achieves the best $F_{\text{max}}$ score 0.768, which demonstrates the necessity of capturing short-, medium- and long-range interactions for protein structure modeling.

## H.3 EFFECT OF GRMP FOR PROTEIN STRUCTURE MODELING

In Tab. 11, we compare between RGConv and GRMP under the comparable throughput (*i.e.*, the number of proteins that the model can process in one second). All experiments are performed on EC with EurNet. (1) We first set the hidden dimension of GRMP as 512. Under the comparable throughput, RGConv can only have the dimension of 422, and its $F_{\text{max}}$ score 0.752 is lower than GRMP's 0.768. (2) We then increase RGConv's hidden

Table 11: Ablation study of multi-relational modeling layer on EC with EurNet.

| Layer | Hidden Dimension | Throughput (proteins/s) | $F_{\text{max}}$ |
|:---|:---:|:---:|:---:|
| RGConv | 422 | 34.4 | 0.752 |
| GRMP | 512 | 34.6 | **0.768** |
| RGConv | 512 | 31.2 | 0.767 |
| GRMP | 592 | 31.5 | **0.780** |

dimension to 512. At this time, RGConv achieves the $F_{\text{max}}$ score 0.767 which is comparable to GRMP's performance under the same dimension, while its throughput is decreased by 3.2. Under the comparable throughput, GRMP can have the hidden dimension of 592, which leads to a higher $F_{\text{max}}$ score 0.780. These results demonstrate that GRMP owns a better efficiency-performance trade-off than RGConv on protein structure modeling.

## I SENSITIVITY ANALYSIS

**Image modeling sensitivity to semantic neighbor size.** In Tab. 12, we report the

Table 12: Sensitivity analysis of semantic neighbor size on ImageNet-1K with EurNet-T.

| #Neighbors | 3 | 6 | 9 | 12 | 15 | 18 | 21 | 24 |
|:---|:---:|:---:|:---:|:---:|:---:|:---:|:---:|:---:|
| Top-1 Acc (%) | 82.23 | 82.22 | 82.16 | 82.26 | 82.20 | **82.34** | 82.28 | **82.34** |

performance of EurNet-T on ImageNet-1K classification under different semantic neighbor sizes for medium-range edge construction. Though some marginal improvements are observed by using a larger neighborhood size (*i.e.*, more than or equal to 18 neighbors), the image modeling performance on this task is in general insensitive to the semantic neighbor size. By default, EurNet-T uses 12 semantic neighbors (denoted by the gray cell in Tab. 12), which achieves comparable performance with the configurations using more semantic neighbors.

