# OpenReview forum: "EurNet: Efficient Multi-Range Relational Modeling of Spatial Multi-Relational Data"
_ICLR.cc/2023/Conference — Submitted to ICLR 2023_

### Official Review · Reviewer_aeee · 2022-10-23

**Confidence:** 2
**Clarity, Quality, Novelty And Reproducibility:** Clarity, Quality, Novelty, and Reprod…
**Correctness:** 3
**Technical Novelty And Significance:** 3
**Empirical Novelty And Significance:** 2
**Recommendation:** 6

**Strength And Weaknesses:**

Strength:
1, Propose a novel modeling layer, called gated relational message passing (GRMP), to propagate multi-relational information across the data.
2, Demonstrate EurNets in image and protein structure modeling cases.
Weaknesses:
1, Why your model is more "efficient" than other models?
2, Why do we need multi-range relational modeling in Image classification?
3, Do you have an inference time comparison between previous works?
4, The method part is somehow hard to follow. Do you have any figures to demonstrate your method?
5, Why relation-channel entangled aggregation is better?

**Summary Of The Paper:**

1, Introduce a task called multi-range relational modeling.
2, Their EurNet constructs the multi-relational graph, where each type of edge corresponds to short-, medium- or long-range spatial interactions.
3, The proposed GRMP separately performs (1) relational message aggregation on each individual feature channel and (2) node-wise aggregation of different feature channels


**Summary Of The Review:**

The paper introduces EurNet to do multi-range relational modeling. The proposed GRMP separately models multi-relational graph.

---

> ### Author Response · Authors · 2022-11-12
> **Author Feedbacks to Reviewer aeee (Part 1/2)**
>
> Thanks for your insightful comments and golden suggestions! We respond to your concerns as below:
>
> >**Q1: Why is your model more "efficient" than other models?**
>
> **The efficiency of our EurNet relies on the efficiency of its main component, the GRMP layer, on modeling multi-relational data.** From the theoretical perspective, compared to the widely-used RGConv layer [a], **the proposed GRMP layer consumes fewer extra floating-point operations (FLOPs) as the increase of the number of relations to be modeled**, as illustrated by the comparison between Equations (2) and (4) and also by the line charts in Figure 1. From the empirical perspective, we study the efficiency of our GRMP against the previous RGConv on image modeling (shown in the second and third lines of Table 6) and protein structure modeling (shown in the second and third lines of Table 11). These two studies show that, for both real-world modeling problems, **GRMP-based models own much higher data throughput (i.e., efficiency) than RGConv-based ones under the same model size and under the comparable model performance.** These evidences verify that, by equipping with GRMP layers, the proposed EurNet is more efficient than the widely-used RGCN [a] on multi-relational data modeling across various domains.
>
> [a] Modeling relational data with graph convolutional networks. Michael Schlichtkrull et al., European Semantic Web Conference, 2018.
>
> >**Q2: Why do we need multi-range relational modeling in image classification?**
>
> This is a good question. We answer it under the support of the ablation results in Table 5. We have the general intuition that, **for image classification tasks that are commonly performed on the global level of images, long-range interaction modeling is most important among short, medium and long ranges**. Such an intuition is verified by the results in the first block of Table 5, in which, when modeling a single spatial range, the model with long-range edges achieves the best performance on ImageNet-1K classification. However, **when the classification relies on fine-grained local interactions, modeling short- and medium-range interactions can also be beneficial**. For example, if we want to classify between cyclists and bicycle repairmen, it is important to model the local interactions between the human and the bicycle. The results in the second and third blocks of Table 5 support this viewpoint, where the performance of the model with only long-range edges is enhanced by including either short-range or medium-range edges, and the highest accuracy is achieved by using all three ranges of edges. **In particular, these performance improvements are obtained by regarding each spatial range as an individual relation and performing multi-relational modeling.** To summarize, supported by these results, we claim the benefits of multi-range relational modeling for image classification.
>
> >**Q3: Do you have an inference time comparison between previous works?**
>
> We study the inference efficiency in terms of data throughput (i.e., the number of samples processed by the model in one second). We compare data throughput between the models based on our proposed GRMP layer and the ones based on the widely-used RGConv layer [a], where the comparison is performed on real-world multi-relational modeling problems of images (shown in the second and third lines of Table 6) and protein structures (shown in the second and third lines of Table 11). In both studies, we have demonstrated that **GRMP-based models own much higher data throughput than RGConv-based ones under the same model size and under the comparable model performance.** These results illustrate that **better inference efficiency can be achieved by using the proposed GRMP layer for multi-relational modeling**.
>
> [a] Modeling relational data with graph convolutional networks. Michael Schlichtkrull et al., European Semantic Web Conference, 2018.
>
> >**Q4: Do you have any figures to demonstrate your method?**
>
> Thanks for this great suggestion! In Appendix C of the revised version, we have supplemented Figure 5 and also corresponding texts to illustrate the proposed GRMP layer more clearly. Please check them out.

---

> > ### Author Response · Authors · 2022-11-12
> > **Author Feedbacks to Reviewer aeee (Part 2/2)**
> >
> > >**Q5: Why relation-channel disentangled aggregation is better?**
> >
> > Here, we illustrate the advantage of the relation-channel disentangled aggregation over the relation-channel entangled aggregation by time complexity analysis. Essentially, both kinds of operations aim to (1) aggregate the information from each node’s neighbors with different relations and (2) aggregate the information from different feature channels. We assume that there are $N$ nodes and $R$ relations in the graph, and the number of feature channels is $C$ both before and after the operation. We respectively illustrate the time complexity of a relation-channel entangled operation and a disentangled one as below:
> >
> > 1. **Relation-channel entangled aggregation:** This kind of operation computes each dimension of output features based on (1) each dimension of input neighborhood features, and also based on (2) the neighborhoods of all relations. Therefore, such an operation has the time complexity of $O(RVC^2)$. **When increasing the number of relations to be modeled, the computational cost will scale up with a factor of $O(VC^2)$, which is prohibitive in many applications with large graphs and large feature size.**
> >
> > 2. **Relation-channel disentangled aggregation:** This kind of operation is performed in a two-step manner. (1) In the first step, each dimension of output features is computed based on all relational neighborhood features at the same dimension, which owns the complexity of $O(RVC)$. (2) In the second step, a typical linear layer is used to fuse the information from all feature channels in a node-wise way, which has the complexity of $O(VC^2)$. Totally, such an operation has the complexity of $O(RVC + VC^2)$, which **scales up with the factor of $O(VC)$ when increasing the relation number, and is thus much computationally cheaper than an entangled operation under large feature size**.
> >
> > Based on these analyses, we conclude that **relation-channel disentangled aggregation can more efficiently model the real-world multi-relational data at scale**. Our proposed GRMP layer is a typical relation-channel disentangled operation, and it shows higher efficiency than the RGConv layer [a] (a typical entangled operation) on both image modeling (shown in the second and third lines of Table 6) and protein structure modeling (shown in the second and third lines of Table 11).
> >
> > [a] Modeling relational data with graph convolutional networks. Michael Schlichtkrull et al., European Semantic Web Conference, 2018.

---

### Official Review · Reviewer_WbJw · 2022-10-23

**Confidence:** 4
**Clarity, Quality, Novelty And Reproducibility:** Clarity, Quality are good, while the …
**Correctness:** 3
**Technical Novelty And Significance:** 3
**Empirical Novelty And Significance:** 2
**Recommendation:** 5

**Strength And Weaknesses:**

Strength
1, The motivation is clear, and the writing is easy to follow.

2, The experiments results are extensive. The proposed approach outperforms previous work FocalNet in many settings.

3, The experiment results for protein function prediction are interesting.

Weakness.
1, The idea or motivation of using different range graph structure is not new before the vision transformer.  The authors do not cite or compare the closely related works: 1, Graph-Based Global Reasoning Networks CVPR-2019. 2, Dynamic Graph Message Passing Network CVPR-2020. 3, Dual Graph Convolutional Network for Semantic Segmentation, BMVC-2019.

2, The core contribution in Equ.3 is not new or novel. It contains the different window sizes of MHSA-like operator and along with a channel attention.

3, The core contribution in Equ.3 should be simplified for easier understanding.  For example, why not give a figure to better illustrate the core operation?

4, What the advantages of combining different range for classification tasks?
What are the advantages over CNN+ transformer-like models?

5, The performance on ImageNet and COCO are not very competitive. In particular, compared with Convnext, the GFlops and Throughput increases while the performance improvements are within 0.2 over different baselines.

Moreover, the proposed framework does not show any performance gains over con-current methods.
CMT: Convolutional Neural Networks Meet Vision Transformers. CVPR-2022
CMT: Convolutional Neural Networks Meet Vision Transformers. Arxiv-2022.02






**Summary Of The Paper:**

This paper introduces the EurNet for Efficient multi-range relational modeling. It constructs the multi-relational graph to encode the short-medium and long-range relation embedding. It also introduces a gated relational message passing layer to achieve that. Extensive experiments on ImageNet classification, COCO object detection and ADE20K semantic segmentation show the effectiveness of proposed approaches.

**Summary Of The Review:**

I appreciate the author present detailed rebuttal. However, the related works I present share the similar ideas with the author work. The authors also admit they ignore these related works. Please do cite these paper for new draft.

After discussion with AC and other reviewers, I keep rating unchanged.

---

> ### Author Response · Authors · 2022-11-12
> **Author Feedbacks to Reviewer WbJw (Part 1/3)**
>
> Thanks for your golden comments and valuable suggestions! We respond to your concerns as below:
>
> >**Q1: Some related works of using different range graph structures are not discussed.**
>
> Thanks for pointing out these important related works! All these works have a common goal as ours, i.e., capturing multi-range spatial interactions, and they achieve such a goal by performing graph convolution over fully-connected graphs [a,b] or dynamic graphs [c] induced from images. The main difference between our method and these previous ones lies in **how to represent the graph**: (1) **previous methods consider a homogeneous graph** where interactions between image regions are represented by a single relation; (2) while **our method employs the multi-relational graph representation** where edges are separated into multiple groups based on different interaction types, which is **a more expressive representation format than the previously-used homogeneous graph**. In the revised paper, we have added the discussion and comparison with these related works to the second paragraph of Section 2.
>
> &emsp;
>
> [a] Graph-based global reasoning networks. Chen et al., CVPR, 2019.
>
> [b] Dual graph convolutional network for semantic segmentation. Zhang et al., BMVC, 2019.
>
> [c] Dynamic graph message passing networks. Zhang et al., CVPR, 2020.
>
> &emsp;
>
> >**Q2: The GRMP layer in Eq. (3) combines MHSA-like operators with different window sizes along with a channel attention, which lacks novelty.**
>
> We first clarify the connections and differences of the GRMP layer against the MHSA-like operator using multiple window sizes. In general, both of them are able to model spatial interactions at different spatial ranges, and the modeling at each spatial range is parameterized individually. However, compared to the MHSA-based operator, **GRMP owns better scalability via sparse connections**. Specifically, MHSA constructs a fully-connected graph in each window, while GRMP sparsely connects interacting nodes at each spatial range. Such sparse connections enable GRMP to easily adapt to large graphs like high resolution images and large protein structures without sacrificing long-range modeling ability. By comparison, the full connection forbids an MHSA-based operator to adopt large enough window sizes to capture all long-range interactions in large graphs. In addition, **when applied to non-Euclidean data like protein structures, the sparse connections of GRMP can help to indicate important geometry-related data properties**, e.g., the functional sites of proteins (these sites always depend on the local density of amino acids, i.e., the node degree). To summarize, we show the key difference of GRMP against an MHSA-based operator, and we illustrate the advantages brought by such different design.
>
> We next clarify the difference between GRMP’s channel-wise relational graph convolution and the conventional channel attention. **The conventional channel attention operates in a point-wise fashion**, where the feature channels at each point/node are independently scaled by attention scores without affecting other points/nodes. However, **the channel-wise relational graph convolution combines channel scaling with message passing**, where the neighborhoods of each node is scaled in a channel-wise way and further aggregated to update the node. Such difference originates from the **different motivations of these two operations**, where **channel attention seeks to perform importance reweighting among feature channels**, while **channel-wise relational graph convolution aims at information propagation across the data**.
>
> **Based on these analyses, we argue that our proposed GRMP layer owns unique and decent contributions.**
>
> >**Q3: A figure is favored to better illustrate the core operations of the GRMP layer in Eq. (3).**
>
> This suggestion is great! In Appendix C of the revised version, we have added Figure 5 and corresponding texts to more clearly illustrate the core operations of the GRMP layer. Please check them out.

---

> > ### Author Response · Authors · 2022-11-12
> > **Author Feedbacks to Reviewer WbJw (Part 2/3)**
> >
> > >**Q4: What are the advantages of combining different ranges for classification tasks?**
> >
> > We answer this question under the support of the ablation results in Table 5. The general intuition is that, **for image classification tasks that are commonly performed on the global level of images, long-range interaction modeling is most important among short, medium and long ranges**. Such an intuition is demonstrated by the results in the first block of Table 5, in which, when a single spatial range is modeled, the model with long-range edges performs best on ImageNet-1K classification. However, **when the classification relies on fine-grained local interactions, modeling short- and medium-range interactions can also help**. For example, if we are to classify between cyclists and bicycle repairmen, it is important to model the local interactions between the human and the bicycle. This viewpoint is supported by the results in the second and third blocks of Table 5, where the performance of the model with only long-range edges is promoted by including either short-range or medium-range edges, and the highest performance is achieved when using all three ranges of edges. These results verify the advantages of combining different spatial ranges for image classification tasks.
> >
> > >**Q5: What are the advantages of EurNet over CNN+Transformer-like models?**
> >
> > Essentially, both the proposed EurNet and the CNN+Transformer-like models [d,e] aim to fully capture different ranges of interactions. CNN+Transformer-like models use convolution to extract short-range interactions and utilize MHSA-based operations to capture long-range interactions, while EurNet attains this goal by relational message passing where each relation corresponds to the interactions at a specific range. Despite the common modeling goal, Eurnet is superior over CNN+Transformer-like models in the following aspects:
> >
> > 1. **More flexible**: CNN+Transformer-like models capture the interactions within the windows of predefined sizes, i.e., kernel size for convolution and patch resolution for MHSA. By comparison, **EurNet involves medium-range edges to capture non-local interactions in an input-adaptive way**, where each node is linked with its most semantically relevant neighbors without the constraint of “windows”. **This flexible linking mechanism allows EurNet to capture non-local interactions based on the structure of input data, instead of in a predefined way as CNN+Transformer-like models.**
> >
> > 2. **More versatile**: The versatility of EurNet stems from its data representation format. Instead of deeming the data as Euclidean-structured ones (e.g., grids or sequences) as CNN+Transformer-like models, **EurNet adopts a non-Euclidean data representation format, i.e., the multi-relational graph, which is more general and can be easily extended to the data from various domains**, including images, protein structures and knowledge graphs. Thanks to such a data representation manner, EurNet can be easily adapted to various domains for solving important problems like image classification, protein function prediction and knowledge graph completion (this task is newly studied and added to Appendix G). **Such versatility cannot be attained by CNN+Transformer-like models.**
> >
> > &emsp;
> >
> > [d] Cvt: Introducing convolutions to vision transformers. Wu et al., ICCV, 2021.
> >
> > [e] Cmt: Convolutional neural networks meet vision transformers. Guo et al., CVPR, 2022.

---

> > > ### Author Response · Authors · 2022-11-12
> > > **Author Feedbacks to Reviewer WbJw (Part 3/3)**
> > >
> > > >**Q6: The performance gains over ConvNeXt [f] are marginal, and the performance of EurNet is comparable to con-current methods like CMT [e].**
> > >
> > > Thank you for extensively comparing across vision backbones. For image modeling, our EurNet is slightly better than ConvNeXt on ImageNet-1K classification in terms of performance and inferior to it in terms of efficiency, while **EurNet outperforms ConvNeXt with a clear margin on ADE20K semantic segmentation**. Compared to CMT, EurNet is inferior on ImageNet-1K classification under the comparable computational cost, while, under the comparable computation, **EurNet-T clearly outperforms CMT-S on COCO object detection and instance segmentation**.
> > >
> > > More importantly, besides several improvements achieved on image modeling, **the core contribution of EurNet is a universal model architecture that can well handle the modeling of spatial multi-relational data and also pure multi-relational data from various domains, including images, protein structures and knowledge graphs.** For protein structure modeling, we demonstrate the superior performance of EurNet over the previous SoTA GearNet [g] (in Section 5.2) and also its superior efficiency-performance trade-off over the widely-used RGCN [h] (in Appendix H.3). For knowledge graph modeling, we prove the substantial performance improvements of EurNet over the previous SoTA CompGCN [i].
> > >
> > > **Based on all these contributions, we argue for the important contributions of our work to the field of general multi-relational data modeling.**
> > >
> > > &emsp;
> > >
> > > [e] Cmt: Convolutional neural networks meet vision transformers. Guo et al., CVPR, 2022.
> > >
> > > [f] A convnet for the 2020s. Liu et al., CVPR, 2022.
> > >
> > > [g] Protein representation learning by geometric structure pretraining. Zhang et al., arXiv:2203.06125 (2022).
> > >
> > > [h] Modeling relational data with graph convolutional networks. Michael Schlichtkrull et al., European Semantic Web Conference, 2018.
> > >
> > > [i] Composition-based multi-relational graph convolutional networks. Vashishth, Shikhar, et al., arXiv:1911.03082 (2019).

---

> ### Author Response · Authors · 2022-12-12
> **Appreciation for pointing out related works**
>
> Thanks for your attention and response to our feedback. We have discussed the three related works [a,b,c] pointed out by you in the revised version, and they will be definitely included in all future versions.
>
>
> &emsp;
>
> [a] Graph-based global reasoning networks. Chen et al., CVPR, 2019.
>
> [b] Dual graph convolutional network for semantic segmentation. Zhang et al., BMVC, 2019.
>
> [c] Dynamic graph message passing networks. Zhang et al., CVPR, 2020.

---

### Official Review · Reviewer_S9ug · 2022-10-24

**Confidence:** 3
**Correctness:** 3
**Technical Novelty And Significance:** 3
**Empirical Novelty And Significance:** 2
**Recommendation:** 6

**Clarity, Quality, Novelty And Reproducibility:**

I'm happy with the clarity, quality and reproducibility of the paper. I have minor concerns on novelty.

**Strength And Weaknesses:**

Strength:

1. The paper provides comprehensive experiments on various tasks of computer vision and protein structure modeling. In each of the tasks, the paper compares with strong baselines on large-scale datasets. Comprehensive ablation study is also provided to show the efficiency of the proposed module. The empirical study of the paper is strong and supports the claims of the paper well.

2. The proposed module shows promising results when applied with different backbone architectures in various tasks, which demonstrates the usability of the method.

3. The paper is well written and structured. Experimental details are elaborated.

Weaknesses:

1. Despite the comprehensive empirical study, I feel that the novelty of the proposed method might be a bit weak. The proposed method is an adaptation from a well studied method (RGConv) from the angle of efficiency. The technical depth might not be very strong. But I would not complain much as similarity might be a factor that the method can be applied in various tasks.

2.  The construction of the graphs may require additional tuning such as the number of KNN. It is also a bit heuristic as it is constructed by some predefined rules (although it can be dynamic with the learning of the model). Moreover, although the paper claims "spatial", I feel that it does not apply specific configurations to deal with "spatial" and "spatial" is just one of the predefined relations.

**Summary Of The Paper:**

This paper introduces a graph convolution module that learns representations of a node by aggregating the representations from its neighbour nodes in a graph. Compared with the standard one (RGConv from Schlichtkrull et al., 2018), the proposed module introduces factorised version of the kernel matrix and other adaptations to reduce the computational complexity. The paper then applied the proposed module to several tasks of computer vision as well as protein structure modeling.

**Summary Of The Review:**

Very comprehensive experiments and novelty is a bit weak.

I have read the response of the authors. Additional experiments are appreciated.

---

> ### Author Response · Authors · 2022-11-12
> **Author Feedbacks to Reviewer S9ug**
>
> Thanks for your constructive comments! We respond to your concerns as below:
>
> >**Q1: The technical depth of the proposed method might not be very strong.**
>
> As you said, we aim at designing **a simple yet effective and efficient multi-relational message passing mechanism (i.e., the GRMP layer) to model the spatial multi-relational data in various domains at scale**. The technical simplicity actually contributes to the general adaptability and effectiveness of the proposed EurNets in various domains.
>
> To study the expressivity of our GRMP layer more in depth, in the Appendix B of the revised version, we theoretically demonstrate that **the expressivity of a GRMP-based model is equivalent to that of the multi-relational version of 1-dimensional Weisfeiler-Leman (1-WL) algorithm [a]**. This result illustrates the power of GRMP on (1) distinguishing the nodes with different structural roles in a multi-relational graph and (2) distinguishing non-isomorphic multi-relational graphs. Please refer to the Appendix B of the revision for more details.
>
> &emsp;
>
> [a] Weisfeiler and leman go neural: Higher-order graph neural networks. Morris et al., AAAI, 2019.
>
> &emsp;
>
> >**Q2: The graph construction is a bit heuristic as it is based on some predefined rules.**
>
> Table A. Sensitivity analysis of semantic neighbor size on ImageNet-1K with EurNet-T.
>
> |#Neighbors|3|6|9|12 (default)|15|18|21|24|
> |:----:|:----:|:----:|:----:|:----:|:----:|:----:|:----:|:----:|
> |Top-1 Acc (%)|82.23|82.22|82.16|82.26|82.20|**82.34**|82.28|**82.34**|
>
> Essentially, we construct different types of edges to separately capture short-, medium- and long-range interactions in the data, and such edge construction is based on the knowledge about interactions at different spatial ranges in a specific domain. We argue that it is important to properly define the interactions at each spatial range, while, **for most hyperparameters in the definitions, we do not need to exhaustively tune them to achieve good performance**.
>
> Here, we study EurNet-T’s sensitivity to the number of semantic neighbors used for modeling medium-range interactions in images, where the performance on ImageNet-1K classification under different semantic neighbor sizes are reported in Table A. Though some marginal improvements are observed by using a larger neighborhood size (i.e., more than or equal to 18 neighbors), the image modeling performance on this task is in general insensitive to the semantic neighbor size. By default, EurNet-T uses 12 semantic neighbors, which achieves comparable performance with the configurations using more semantic neighbors. This sensitivity analysis has been added to the Appendix I of the revision. **We will finish analyzing the sensitivity of all other graph construction hyperparameters and add the full analysis to the next version.**
>
> >**Q3: No specific configurations are applied to deal with “spatial”, and “spatial” is just one of the predefined relations.**
>
> We would like to clarify the philosophy of our model design: **(1) we put all designs of spatial modeling in multi-range relational graph construction, which is led by domain experts; (2) we make the encoder part domain- and application-agnostic, such that machine learning practitioners can get rid of the heavy work of designing domain-specific spatial encoding mechanisms.** In fact, the spatial patterns of data are quite different across various domains. For example, image patches lie in the 2D space, while amino acids of a protein lie in the 3D space; image patches are located by discrete 2D coordinates, while amino acids are located by continuous 3D positions, etc. Therefore, it is arduous for machine learning practitioners if they have to re-design an encoder every time facing a spatial modeling problem in a new domain. By comparison, **it could be much more efficient to let domain experts lead the design of edge construction for spatial modeling and let machine learning practitioners focus on optimizing a better model with a domain-agnostic encoder.** **We follow such a philosophy to design the proposed EurNets.**

---

### Official Review · Reviewer_nyUd · 2022-11-02

**Confidence:** 3
**Correctness:** 4
**Technical Novelty And Significance:** 3
**Empirical Novelty And Significance:** 3
**Recommendation:** 8

**Clarity, Quality, Novelty And Reproducibility:**

* Clarity: clearly written
* Novelty: Novel. New layer for multi-relational modelling over graphs at scale.
* Reproducibility: Good. Hyperparameter choices are well justified. Criteria on the choice of baseline models are discussed. Authors provided detailed architectures of the proposed models. Source code in supplementary material.


**Strength And Weaknesses:**

Strengths:
* Contributions are clear and important
* Claims are well supported; experiments are extensive (compared with multiple methods, using different model capacities, and considered different datasets / tasks)
* The paper is well written. Figures are clarifying.

Weakness:
* This is a minor point: Maybe the authors could more explicitly showcase the empirical advantages of GRMP speedup in an applied setting when comparing with graph-based multi-relational models. (this is somehow already discussed in the “Throughput analysis.” paragraph, but the authors could make it more evident)

**Summary Of The Paper:**

In this paper, the authors introduce a novel layer, called Gated Relational Message Passing (GRMP), for efficient multi-relational modelling over graphs. Importantly, GRMP scales better than existing multi-relational models when increasing the number of considered relations.

The authors use GRMP as building block for Efficient multi-range relational graph neural networks (EurNet) that they apply to multi-scale modelling problems. Across image classification, object detection, image segmentation and protein function prediction tasks, the authors show that EurNet performs comparably or better than state-of-the-art methods.

**Summary Of The Review:**

This is an important contribution to multi-relational modelling and graph neural networks. The paper is clearly written, experiments are extensive and detailed supplementary info is provided.

---

> ### Author Response · Authors · 2022-11-12
> **Author Feedbacks to Reviewer nyUd**
>
> Thanks for your appreciation of our work! We respond to your concern as below:
>
> >**Q1: Authors could more explicitly showcase the empirical advantages of GRMP speedup in an applied setting.**
>
> This advice is great. In the revised version (last sentence of Section 3.3), we have explicitly pointed out two empirical studies about GRMP’s efficiency for image modeling (the second paragraph of Section 5.3) and protein structure modeling (Appendix H.3). These two studies show that, for both real-world modeling problems, GRMP-based models own higher data throughput (i.e., efficiency) than RGConv-based ones under the same model size. In addition, by adjusting the model size to endow the models with comparable data throughput, GRMP-based models achieve apparently higher performance than RGConv-based ones. **These results demonstrate the better efficiency-performance trade-off gained by the proposed GRMP in applied settings.**

---

### Author Response · Authors · 2022-11-12
**Summary of Response**

We would like to thank all reviewers for your time and constructive suggestions in our paper!

We have posted the responses to your questions and revised the paper for better theoretical and empirical guarantees, where **the revisions are marked in RED** (only the section name is in red if the whole section is newly added). Here is a brief summary of important points:

1. **The novelty concern (Reviewer S9ug, WbJw):** In the Appendix B of the revision, to understand the proposed GRMP layer more deeply, we have added a theoretical analysis about its expressivity. Also, we give point-by-point comparisons between our model and previous related ones (multi-head self-attention, channel attention, CNN+Transformer-like models) in the responses to Reviewer WbJw. In summary, we illustrate the novelty of our method theoretically and conceptually.

2. **More intuitive illustration of the proposed method (Reviewer WbJw, aeee)**: In the Appendix C of the revised paper, we have added Figure 5 and also corresponding texts to illustrate the proposed GRMP layer more clearly.

3. **Empirical efficiency and effectiveness (Reviewer nyUd, WbJw, aeee):** For empirical efficiency, in Table 6 and 11, we have demonstrated the superior inference throughput and efficiency-performance trade-off of our GRMP layer over the widely-used RGConv layer on image and protein structure modeling problems. For the general effectiveness on modeling spatial multi-relational data and pure multi-relational data, in the Appendix G of the revision, we have newly added the results on knowledge graph reasoning tasks and verified the superior performance of our EurNet on these tasks. Together with the results on image and protein structure modeling tasks, we claim the general effectiveness of our approach on multi-relational modeling at scale across various domains.

---

### Decision · Program_Chairs · 2023-01-20

**Decision:**

Reject

**Justification For Why Not Higher Score:**

My major concerns lie in presentation, motivation, and philosophy, which makes this paper not self-standing and lacks novelty. Two reviewers also have concerns on the novelty, but they would like to insist their scores.

**Justification For Why Not Lower Score:**

Although this paper receives four positive scores, several reviewers have concerns on the novelty. Moreover, it is weird to see that one reviewer increased the score due to its limited novelty. I asked the reason but did not receive any reply from the reviewer.

Based on my above comments, this paper does not reach the bar of ICLR.

**Metareview: Summary, Strengths And Weaknesses:**

Based on the collected information from all reviewers and my personal judgement, I can make the recommendation on this paper, **rejection**. Here are the comments that I summarized, which include my opinion and evidence.

**Research Problem**

This paper studies the spatial relationship in various learning tasks including image classification and protein structure.

**Related Work**

Several reviewers pointed out that many key missing references.

**Motivation**

The authors argue that there exist different kinds of spatial relations, including short-range, medium-range, and long-range relations. Only involving a single type of spatial relations is not sufficient for the learning task. However, as the introduction says, there exists a branch of literature in multi-relational modeling. This motivation is not strong. Moreover, I am also confused about “separately modeling each spatial relation is a better solution to capture different tasks’ focus.” Why? More explanations and support are needed.

**Philosophy**

The authors aim to propose a new method, named EurNet, for multi-range relational modeling. From the above motivation, I am not informed or confused about what challenges the authors aim to tackle. Therefore, it is unclear the philosophy to handle these challenges. As the authors mentioned the great importance of short-range relations in instance segmentation, and the upgraded importance of long-range relations in semantic segmentation, relational learning is task-dependent. Separately modeling each spatial relation will not solve the problem. The authors fail to answer what type of spatial relation is suitable for the learning task, or how to fuse different types of spatial relations for the learning task.

**Presentation**

The presentation is difficult to follow and should be improved heavily. I have read this manuscript at least five times. Many parts are prolix and lack information. The authors might have a misunderstanding on problem definition in Section 3.1. The problem definition is the research problem that the authors aim to address, rather than the proposed model to tackle the research problem.  A formal problem definition is strongly suggested with clear notations

**Technique**

The authors propose multi-range relational edge construction and dynamic edge construction with two instantiations in image and protein structure modeling. However, the parameter settings in Section 4 are ad-hoc.

**Experiments**

(1) The experimental results are extensive.

(2) I guess that the performance gain comes from multiple types of inputs, which can be verified by the ablation study in Table 5.

(3) The authors should also feed the multiple types of inputs for different message-passing strategies. By this means, the authors can demonstrate the effectiveness of their proposed GRMP in a fair fashion.

No objection was raised from the reviewers on the rejection recommendation.


**Summary Of Ac-Reviewer Meeting:**

This is not a borderline paper. One reviewer is reluctant to launch the zoom meeting.